# MALinZero: Efficient Low-Dimensional Search for Mastering Complex Multi-Agent Planning

**Sizhe Tang**
The George Washington University
`s.tang1@gwu.edu`

**Jiayu Chen**
Carnegie Mellon University
`jiayuc2@andrew.cmu.edu`

**Tian Lan**
The George Washington University
`tlan@gwu.edu`

## Abstract

Monte Carlo Tree Search (MCTS), which leverages Upper Confidence Bound for Trees (UCTs) to balance exploration and exploitation through randomized sampling, is instrumental to solving complex planning problems. However, for multi-agent planning, MCTS is confronted with a large combinatorial action space that often grows exponentially with the number of agents. As a result, the branching factor of MCTS during tree expansion also increases exponentially, making it very difficult to efficiently explore and exploit during tree search. To this end, we propose MALinZero, a new approach to leverage low-dimensional representational structures on joint-action returns and enable efficient MCTS in complex multi-agent planning. Our solution can be viewed as projecting the joint-action returns into the low-dimensional space representable using a contextual linear bandit problem formulation. We solve the contextual linear bandit problem with convex and $\mu$-smooth loss functions – in order to place more importance on better joint actions and mitigate potential representational limitations – and derive a linear Upper Confidence Bound applied to trees (LinUCT) to enable novel multi-agent exploration and exploitation in the low-dimensional space. We analyze the regret of MALinZero for low-dimensional reward functions and propose an $(1 - \frac{1}{e})$-approximation algorithm for the joint action selection by maximizing a sub-modular objective. MALinZero demonstrates state-of-the-art performance on multi-agent benchmarks such as matrix games, SMAC, and SMACv2, outperforming both model-based and model-free multi-agent reinforcement learning baselines with faster learning speed and better performance.

## 1 Introduction

Monte Carlo Tree Search (MCTS) has demonstrated great performance in solving complex planning problems such as game playing [1], robotic control [2], and optimization [3]. It achieves much higher data efficiency than value- or policy-based reinforcement learning (RL) [4] by leveraging Upper Confidence Bound for Trees (UCTs) to balance exploration and exploitation through randomized sampling and cumulative regret minimization [5]. Integrated with deep learning (e.g., AlphaZero [6] and MuZero [4]), MCTS algorithms have achieved groundbreaking results in solving complex games, such as Go, Chess and Shogi [4], relying on little knowledge of domain expertise or game rules.

However, for planning problems involving multiple agents, MCTS is confronted with a large combinatorial action space that often grows exponentially with the number of agents [7, 8]. As the number of candidate actions increases, the branching factor of MCTS (during tree expansion) also increases

39th Conference on Neural Information Processing Systems (NeurIPS 2025).

exponentially, making it very difficult to efficiently explore and exploit during tree search [9, 10]. Existing works either focus on single-agent problems or limit tree search to a small set of state-conditioned action abstractions [9, 11]. As a result, MCTS can get stuck in local optima or become slow to explore optimal actions. Recent proposals like MAZero [10] facilitate distributed representation of state transitions and reward prediction in multi-agent MCTS, but again do not address the challenges relating to the combinatorial action space in multi-agent planning.

In this work, we propose MALinZero, a new approach to leverage low-dimensional representational structures and enable efficient MCTS in complex cooperative multi-agent planning. The main idea of MALinZero is to model the joint returns through a low-dimensional linear combination of the (latent) per-agent action rewards. Thus, by observing the joint returns resulted from multi-agent actions, we can formulate a contextual linear bandit problem [12] – with the per-agent action rewards as an unknown parameter vector $\theta$ – and derive a linear Upper Confidence Bound applied to trees (LinUCT), to enable novel LinUCT-based exploration and exploitation in this low-dimensional space of (latent) per-agent action rewards. The idea of enforcing representational structures on joint returns has been instrumental in multi-agent reinforcement learning (MARL), e.g., VDN [13] with linear representations, and QMIX [14], NDQ [15], and PAC [16] with monotonic representations, as well as policy factorizations like DOP [17] and FOP [18]. However, these MARL results do not apply to multi-agent MCTS, which requires not only factorized action-values but also the use of concentration inequalities [19] to bound their probability distributions given observed samples, like in our LinUCT.

For a planning problem with $n$ agents and $d$ actions per agent, MALinZero effectively reduces the tree search from considering $d^n$ independent joint-action returns to learning $nd$ latent per-agent action rewards. Our solution can be viewed as projecting the returns into the low-dimensional space represented by MALinZero using a contextual linear bandit problem formulation [12, 20]. To mitigate the potential representational limitations, we further introduce a strongly-convex, $\mu$-smooth distance measure $f$ into the projection (as a new contextual bandit loss), in order to place more importance on not underestimating the better joint actions, while not overestimating the less attractive joint actions [21]. We solve the resulting contextual linear bandit problem with this convex loss and prove that our LinUCT achieves an cumulative regret of $\hat{R}_T = O\big(nd \cdot \sqrt{\mu T} \cdot \ln(T)\big)$ after $T$ steps for low-dimensional rewards. We further show that the joint action selection problem in our MALinZero is a maximization of a submodular objective and can be solved using an $(1 - \frac{1}{e})$-approximation algorithm. MALinZero achieves state-of-the-art performance in our evaluations on matrix games, SMAC [22], and SMACv2 [23], by enabling multi-agents MSCT via low-dimensional representations.

The primary contributions of this paper are as follows:

- We propose MALinZero to leverage low-dimensional representational structures on joint-action returns and enable efficient MCTS in complex multi-agent planning.

- We solve the resulting contextual linear bandit problem with a convex loss function and derive a novel LinUCT to facilitate exploration and exploitation in low-dimensional space.

- We analyze the regret of MALinZero for low-dimensional rewards and proposes an $(1 - \frac{1}{e})$-approximation algorithm for joint action selection via a submodular maximization.

- MALinZero demonstrates state-of-the-art performance on multi-agent planning benchmarks such as MatGame, SMAC, and SMACv2, outperforming both multi-agent RL and MCTS baselines in terms of faster learning speed and better performance.

## 2 Related Works and Background

Multi-agent planning with joint rewards can be modeled as a Decentralized Partially Observable Markov Decision Process (Dec-POMDP), as a tuple $(\mathcal{I}, \mathcal{S}, \{\mathcal{A}\}_{i \in \mathcal{I}}, P, R, \{\Omega\}_{i \in \mathcal{I}}, \{\mathcal{O}\}_{i \in \mathcal{I}}, \gamma)$ [24], where $\mathcal{I} = 1, 2, \ldots, n$ is the set of $n$ agents, $\mathcal{S}$ the global state space, $\mathcal{A}_i$ the action space of agent $i$, $P$ the state transition probability distribution, $R$ the joint reward function, $\Omega_i$ the individual observation space of agent $i$, $\mathcal{O}$ the global observation function and $\gamma$ the discount factor to weigh future rewards. At times step $t$, agent $i$ gets state $s_t$ thus acquiring local observation $o_t^i = \mathcal{O}^i(s_t)$, then chooses action $a_t \in \mathcal{A}_i$ based on the acquired local observation $o_{\leq t}^i$. Given a joint action $\mathbf{a}_t = \big(a_t^1, \ldots, a_t^N\big)$, the environment transits to the next state $s_{t+1}$ and returns a reward $r = R(s_t, \mathbf{a}_t)$. Agents aim to learn a joint policy $\boldsymbol{\pi}$ that maximizes the expectation of discounted return $E_{\boldsymbol{\pi}}\big[\sum_{t=0}^{\infty} \gamma^t r_t | a_t^i \sim \pi_t^i(\cdot|o_{\leq t}^i), i = 1, \ldots, N\big]$.

**MARL with factorized representations.** Factorization-based methods have been commonly used to cope with the exponentially growing joint state-action space in MARL. Under the notion of Centralized Training and Decentralized Execution (CTDE), algorithms like VDN [13] learn a centralized joint action-value function $Q_{\text{tot}}$ through a linear combination of local per-agent value functions. This is further extended to monotonic representations in QMIX [14], nearly decentralized representations in NDQ [15], and counterfactual predictions in PAC [16]. Policy-based factorizations have also been considered in DOP [17] and FOP [18]. To mitigate potential representation limitation, QTRAN [25] has considered adding state-value correction terms, while Weighted QMIX [14] introduces importance weights on dominant state-actions. The idea of enforcing these representational structures has been instrumental in developing decentralized, scalable MARL algorithms. However, these factorized representations in MARL do not apply to multi-agent MCTS, which requires the use of concentration inequalities to bound the return distributions given observed samples, in order to balance exploration and exploitation.

**MCTS-based planning.** MCTS is widely applied to solve planning problems through sequential decision-making [26]. Efficient search for optimal actions in a large decision space has been one of the central problems in MCTS [27, 10, 4]. Existing works have leveraged Boltzmann policies [11] and state-conditioned action abstractions [9]. The problem becomes more pronounced in multi-agent MCTS, as the joint action space increases exponentially as the number of agents grows [28, 29, 30], leading to significantly increased complexity in tree expansion and search. Recent approaches like MAZero [10] have considered multi-agent MCTS, but only considered distributed representation of state transitions and reward prediction, without addressing the combinatorial action space in multi-agent planning.

MCTS typically involves four stages, i.e., *Selection* to choose actions using UCB-like strategies [31], *Expansion* to add new child nodes, *Simulation* to sample payoffs, and *Back-Propagation* to propagate payoffs and update node returns. Model-based MCTS algorithms like MuZero [4] learn a dynamic model to replace *Simulation*, thus improving the planning efficiency. MuZero involves three key learnable models: a representation model $h_\theta$ to map the real environment into a latent space, a dynamics model $g_\theta$ that computes the next state and the reward of this transition, and a prediction model $f_\theta$ for value and policy approximation. Given the observation history $\mathbf{o}_{<t}$ at time step $t$, the model maps the observation into a latent space as $\mathbf{s}_{t,0} = h_\theta(\mathbf{o}_{<t})$, then unrolls $K$ steps and predicts the corresponding $\mathbf{s}_{t,k}, r_{t,k} = g_\theta(\mathbf{s}_{t,k-1})$ and $v_{t,k}, \mathbf{p}_{t,k} = f_\theta(s_{t,k})$ for each hypothetical step $k$ with $k = 0, 1, \ldots, K$. During *Selection*, MuZero traverses from the root node and applies the probabilistic Upper Confidence Tree (pUCT) rule to select actions for node transitions until reaching the leaf node of the current tree:

$$a = \arg\max_{a \in \mathcal{A}} \quad \Phi(s, a) + c(s) P(s, a) \frac{\sqrt{\sum_b N(s, b)}}{N(s, a) + 1} \tag{1}$$

where $s$, $a$ and $\mathcal{A}$ are abbreviations for $\mathbf{s}_{t,k-1}, a_{t,k}$ and action set respectively. $\Phi(s, a)$ is the estimation for the real value of nodes, $N(s, a)$ denotes the visiting count, $P(s, a)$ is the prior probability of selecting $a$ in $s$, and $c(s)$ is the coefficient balance exploitation and exploration. When the leaf node is reached, new nodes will be expanded to the tree, then $\Phi(s, a)$ and $N(s, a)$ of nodes in the search path will be updated. Specifically, $\Phi(s, a)$ is updated based on a cumulative discounted reward $G_{t,k} = \sum_{\tau=0}^{l-1-k} \gamma^\tau r_{k+1+\tau} + \gamma^{l-k} v^l$ for $k = 0, 1, \ldots, l$ where $l$ is the search depth and thus calculated as $\Phi(s, a) = \frac{N(s,a) \cdot \Phi(s,a) + G_{t,k}}{N(s,a) + 1}$.

Sampled MuZero [32] extends MuZero into a sampling-based framework to tackle larger action spaces for which MuZero can not construct all possible states as nodes. In *Expansion*, only a subset $T(s)$ of the complete action space will be considered by Sampled MuZero according to the sampling policy $\beta$ and prior policy $\pi$. Then the sampled action will be selected by $a = \arg\max_{a \in \mathcal{A}} \quad \Phi(s, a) + c(s) \frac{\hat{\beta}}{\beta} P(s, a) \frac{\sqrt{\sum_b N(s,b)}}{N(s,a)+1}$, where $\hat{\beta}$ is the empirical action distribution.

## 3 MALinZero for Multi-Agent MCTS

MALinZero leverages low-dimensional representations of the joint-action returns and solves the resulting contextual linear bandit problem to enable efficient LinUCT-based MCTS in complex multi-agent planning. LinUCT is applied in *Selection* described in Section 2 to choose the optimal action during MCTS. MALinZero consists of four main modules: the representation model for obtaining

the latent per-agent action rewards as an unknown parameter vector $\theta$ from observed samples, the dynamics model for predicting the next latent state and reward, the prediction model for estimating the search policy and action-values, and the communication model for describing the coordination among multi-agents[1]. Notably, the proposed LinUCT-based search and dynamic node generation in MALinZero would not incur any extra neural networks compared with MAZero, since they depend only on the inner process of each rollout. We analyze the regret of MALinZero for low-dimensional rewards. For action selections, we will show that the problem is a maximization of a sub-modular objective, solvable by an $(1 - \frac{1}{e})$-approximation algorithm. All proofs are collected in the Appendix.

## 3.1 Leveraging Low-Dimensional Representations

MALinZero models the joint-action returns through a low-dimensional linear combination of the latent per-agent action rewards. More precisely, we consider a contextual linear bandit problem [20] with a finite joint-action set $\mathcal{A} \subset \mathbb{R}^{nd}$, where we assume that each agent has $d = |\mathcal{A}_i|$ actions without loss of generality. Thus, each joint action $a \in \mathcal{A}$ is represented by an $n$-hot vector selecting one local action for each agent. It is easy to see that the Euclidean norm of any action is bounded by $\|a\|_2 \leq L = \sqrt{n}, \forall a \in \mathcal{A}$. At each round $t$, we chooses an action $A_t \in \mathcal{A}$, and the environment reveals a reward $X_t = R(s_t, A_t)$.

In this work, we leverage a low-dimensional representation of the reward, i.e., $X_t = \langle \theta^*, A_t \rangle + \varepsilon_t$. Here $\varepsilon_t$ is conditionally $1-$subgaussian[2] observation noise, and $\theta^* \in \mathbb{R}^{nd}$ is an unknown parameter vector representing the (latent) per-agent action return values. Thus, for each $n$-hot vector action $A_t \in \mathcal{A}$, we model the low-dimensional reward $X_t$ as a linear sum of $n$ corresponding per-agent action rewards. Our model can be viewed as projecting the reward $R(s_t, A_t)$ into the low-dimensional space representable using $X_t = \langle \theta^*, A_t \rangle + \varepsilon_t$. It reduces the MCTS from considering $d^n$ joint reward values in each state $s_t$ to learning an unknown parameter vector of size $nd$ only, thus allowing quick estimate of the global reward structure from limited samples and significantly speed-up the tree search in multi-agent MCTS. Applying the regularized least-squares estimator, we can get the empirical estimation of $\theta^*$ from observed samples $X_1, \ldots, X_t$ as

$$\hat{\theta}_t = \arg \min_{\theta \in \mathbb{R}^{nd}} F_t(\theta), \text{ s.t. } F_t(\theta) = \sum_{s=1}^t f(X_s - \langle \theta, A_s \rangle) + \frac{\lambda}{2} \|\theta\|^2 \tag{2}$$

where $f$ is some distance measure, $\|\theta\|^2$ a regularization term ensuring the uniqueness of the solution, $\lambda$ an appropriate constant for the regularization term.

**Classic LinUCB for Euclidean distance $f$.** When $f$ is the Euclidean distance measure, the solution to the estimation problem in (2) can be obtained by differentiation, i.e., $\hat{\theta}_t = V_t^{-1} \sum_{s=1}^t A_s X_s$ where $V_t$ are $nd \times nd$ matrices given by $V_0 = \lambda I$ and $V_t = V_0 + \sum_{s=1}^t A_s A_s^\top$. We can then apply the Upper Confidence Bound (UCB) algorithm [31] to seek the optimal action of stochastic linear bandits, which implements the "optimism in the face of uncertainty" principle. Let $\text{UCB}_t(a) = \max_{\theta \in \mathcal{C}_t} \langle \theta, a \rangle$ be an upper bound on the mean payoff $\langle \theta^*, a \rangle$ for action $a \in \mathbb{R}^{nd}$ where $\mathcal{C}_t \subseteq \mathbb{R}^{nd}$ is the confidence set based on the action-reward history that contains the unknown $\theta^*$ with high probability. At each time $t$, LinUCB [20] selects $A_t = \arg \max_{a \in \mathcal{A}} \text{UCB}_t(a)$. The cumulative regret after $T$ steps is bounded by $R_T = \sum_{t=1}^T (\langle A^*, \theta^* \rangle - \langle A_t, \theta^* \rangle) \leq Cnd\sqrt{T} \log(T\sqrt{n})$ where $A^* = \arg \max_{a \in \mathcal{A}} \langle a, \theta^* \rangle$, and $C > 0$ is a constant.

**Mitigating representational limitations with more general $f$.** While classic bandit algorithms like UCB1 and LinUCB [20] solve the contextual linear bandit problem with Euclidean distance $f$, it does not necessarily yield the best model in terms of exploring the optimal actions in MCTS. Intuitively, the use of low-dimensional representation of the reward may introduce potential representational limitations, as previously observed in MARL algorithms like Weighted QMIX [21]. To explore the optimal actions in MCTS, it is important not to underestimate the better joint actions, while not to

---

[1]Due to space limitation, the specific model architecture can be found in Appendix B.

[2]A random variable $X$ is 1-subgaussian if it satisfies the moment generating function bound $\mathbb{E}[e^{\lambda X}] \leq e^{\lambda^2/2}$ for all $\lambda \in \mathbb{R}$. This implies rapid tail decay $\mathbb{P}(|X| \geq t) \leq 2e^{-t^2/2}$, analogous to a Gaussian with unit variance. The property is central to deriving sharp concentration bounds in statistical learning theory.

overestimate the less attractive ones – which otherwise may lead to substantial errors in recovering the correct maximal actions.

To this end, we consider a general family of strongly-convex, $\mu$-smooth distance measure $f$ in the contextual linear bandit problem in (2). For higher observed rewards $X_t$ that are likely optimal, the distance measure $f$ will have a larger acceleration (i.e., second order derivative if differentiable) for underestimating $(X_s - \langle \theta, A_s \rangle) > 0$, while having a smaller acceleration for overestimating $(X_s - \langle \theta, A_s \rangle) < 0$. On the other hand, for higher observed rewards $X_t$ that are unlikely to be chosen, it is important not to overestimate by having a larger acceleration for $(X_s - \langle \theta, A_s \rangle) < 0$. An example of such $f$ is to consider: $f(X_s - \langle \theta, A_s \rangle) = w_+ \cdot (X_s - \langle \theta, A_s \rangle)^2$ if $X_s \geq \langle \theta, A_s \rangle$, and $f(X_s - \langle \theta, A_s \rangle) = w_- \cdot (X_s - \langle \theta, A_s \rangle)^2$ otherwise. We can choose $w_+ > w_-$ for better $X_s$ to prevent underestimation and $w_+ < w_-$ for undesirable $X_s$. This ensures that our low-dimensional representation in MALinZero can best support the exploration of the optimal actions in MCTS. We will drive a novel LinUCT with respect to such $f$ and leverage it to balance exploration and exploitation in MCTS.

### 3.2   Deriving LinUCT and Analyzing Regret

We derive action selection using LinUCT in MALinZero and provide a cumulative regret bound for the resulting contextual linear bandit problem, depending on the properties of strongly-convex, $\mu$-smooth $f$. We prove that LinUCT can achieve an regret of $\hat{R}_T = O(nd \cdot \sqrt{\mu T} \cdot \ln(T))$ after $T$ steps, ensuring the exploration efficiency using LinUCT. Our analysis builds upon [33] and extends it to general convex loss $f$.

Let $\{A_t\}_{t=1}^T \subset \mathbb{R}^{nd}$ be a sequence of action vectors with $\|A_t\|_2 \leq \sqrt{n}$, and suppose the observed reward at time $t$ is $X_t = \langle \theta^*, A_t \rangle + \eta_t$ where $\theta^* \in \mathbb{R}^{nd}$ satisfies $\|\theta^*\|_2 \leq S$ for some bound $S$, and each $\eta_t$ is conditionally 1-subgaussian. Since $f : \mathbb{R} \to \mathbb{R}$ is strongly-convex and $\mu$-smooth, we have $\varepsilon \leq f''(z) \leq \mu, \forall z \in \mathbb{R}$ for some positive $\varepsilon$. The solution to (2) is obtained by differentiation and yields $\hat{\theta}_t = V_t^{-1} \sum_{s=1}^t w_s A_s X_s$ where we use $w_t = f''(\xi_t)$ with $\xi_t \in (0, X_t - \langle \theta_{t-1}, A_t \rangle)$ and thus have $\varepsilon \leq w_t \leq \mu$ for any $t$ and $\xi_t$. Here $X_s$ is the immediate reward at step $t$. Further, $V_t$ are $nd \times nd$ matrices given by initial $V_0 = \lambda I$ for some constant $\lambda > 0$ and $V_t = V_0 + \sum_{s=1}^t w_s A_s X_s$.

Next, we consider an ellipsoid confidence set centered around the optimal estimator $\hat{\theta}_{t-1}$, i.e., $\mathcal{C}_t = \left\{ \theta \in \mathbb{R}^{nd} : \|\theta - \hat{\theta}_{t-1}\|_{V_{t-1}} \right\} \leq \beta_t$, for an increasing sequence of $\beta_t$ with $\beta_1 \geq 1$ [33]. Note that as $t$ grows, this ellipse $\mathcal{C}_t$ is shrinking as $V_t$ has increasing eigenvalues and if $\beta_t$ does not grow too fast. We show that the problem of selecting optimal action $A_t \in \mathcal{A}$ by solving $\max_{A_t \in \mathcal{A}, \theta \in \mathcal{C}_t} \langle \theta, a \rangle$ in this contextual linear bandit problem is equivalent to:

$$A_t = \arg\max_a \left\langle \hat{\theta}_{t-1}, a \right\rangle + \beta_{t-1} \|a\|_{V_{t-1}^{-1}},$$

which is referred to as our LinUCT rule for action selection. We consider the realized regret defined by $\hat{R}_T = \sum_{t=1}^T (X_t^* - X_t) = \sum_{t=1}^T (\langle \theta^*, A_t^* \rangle - \langle \theta^*, A_t \rangle) + \sum_{t=1}^T (\eta_t^* - \eta_t)$. The next theorem gives the regret bound of LinUCT, with corresponding proofs in Appendix A.

**Theorem 1.** *[Regret Bound of LinUCT] With probability* $1 - \delta$, *the regret of LinUCT satisfies*

$$\hat{R}_t \leq \sqrt{8\mu t \beta_t \ln\left(\frac{\det(V_t)}{\det(\lambda I)}\right)} \leq \sqrt{8\mu nd t \beta_t \ln\left(\frac{nd\lambda + \mu nt}{nd\lambda}\right)}. \tag{3}$$

**Proof sketch**   Let $S_t = \sum_{s=1}^t w_s A_s \eta_s$ and $V_t = \lambda I + \sum_{s=1}^t w_s A_s A_s^\top$. (i) A standard self–normalized concentration (mixture supermartingale) gives, for all $t \leq T$ with probability $\geq 1 - \delta$,

$$S_t^\top V_t^{-1} S_t \leq 2\mu \ln\left(\frac{\det(V_t)^{1/2}}{\lambda^{nd/2}\delta}\right) \quad \Rightarrow \quad \|\hat{\theta}_t - \theta^*\|_{V_t} \leq \beta_t. \tag{4}$$

(ii) By optimism of LinUCT and the confidence event,

$$r_t := X_t^* - X_t \leq \beta_{t-1}\|A_t\|_{V_{t-1}^{-1}} + \Delta_t, \qquad \Delta_t := \eta_t^* - \eta_t. \tag{5}$$

Since $\eta_t, \eta_t^*$ are 1-sub-Gaussian, $\sum_{t=1}^T \Delta_t \leq 2\sqrt{T \ln(1/\delta)}$ w.p. $\geq 1 - \delta$.

(iii) Summing and applying Cauchy–Schwarz plus the (weighted) elliptical potential lemma,

$$\sum_{t=1}^{T} \beta_{t-1} \|A_t\|_{V_{t-1}^{-1}} \leq \sqrt{T} \, \beta_T \sqrt{2 \ln\left(\frac{\det(V_T)}{\lambda^{nd}}\right)}, \tag{6}$$

which, together with (ii) and the definition of $\beta_T$, yields

$$\widehat{R}_T \leq \sqrt{8\mu \, T \, \beta_T \, \ln\left(\frac{\det(V_T)}{\lambda^{nd}}\right)}. \tag{7}$$

(iv) Using $w_t \leq \mu$ and $\|A_t\|_2 \leq \sqrt{n}$, $V_T \preceq \lambda I + \mu n T \, I$, hence

$$\ln\left(\frac{\det(V_T)}{\lambda^{nd}}\right) \leq nd \, \ln\left(\frac{nd\lambda + \mu n T}{nd\lambda}\right), \tag{8}$$

giving the displayed bound in the theorem.

Choosing $\beta_t = \sqrt{2\mu \, \ln\left(\frac{\det(V_t)^{1/2}}{\det(\lambda I)^{1/2} \, \delta}\right)} + \sqrt{\lambda} \, S$, we show that the regret has the following order:

**Corollary 2** (The Order of Regret Bound for LinUCT). *Under the above conditions, the cumulative regret bound of LinUCT with $\delta = 1/T$ satisfies*

$$\hat{R}_T = O\left(nd \cdot \sqrt{\mu T} \cdot \ln(T)\right). \tag{9}$$

The regret bound of LinUCT in Theorem 1 only depends on $nd$ rather than the exponential size of the joint action space. The general convex loss $f$ incurs an extra multiplicative factor $\sqrt{\mu}$ compared with the standard results of contextual bandit [33].

### 3.3 Dynamic Node Generation

MALinZero allows modeling the joint action space using low-dimensional representation, thus significantly speeding up exploration and exploitation in multi-agent MCTS. Specifically, when the leaf node $\Upsilon$ in the search path is visited for the first time, $\kappa = \zeta\chi$ nodes will be sampled as child nodes where $\zeta$ is the dynamic generation ratio and $\chi$ is the maximum number of child nodes. In the subsequent *Selection* stage, node $\Upsilon$ will utilize the cumulative $\theta$ and $V$ (We omit the subscript $t$ in this section for abbreviated notations) to search for the potential optimal action from the entire joint action space and add it as the new child node. If there is no node with a higher value, *Selection* will sample and compare the existing ones. The detailed process can be found in Algorithm 1.

For a root or leaf node $\Upsilon$, $\kappa = \zeta\chi$ nodes are sampled for initialization similar to MAZero. The next time $\Upsilon$ is visited, MALinZero selects optimal action using LinUCT with search policy $P(s, a)$:

$$a = \arg\max_{a \in \mathcal{A}} \Psi(a) = \arg\max_{a \in \mathcal{A}} \, a^\top \theta + c(s) P(s, a) \, \mathrm{trace}(V) \sqrt{a^\top V^{-1} a} \tag{10}$$

where $c(s)$ is a constant, $\Psi(a)$ is the objective function for action selection, and $P(s, a)$ is the search policy used as a prior information in LinUCT similar to MuZero [4]. If the selected action for which the corresponding node does not exist, this node is added after *Selection*. Once a node has $\chi$ child nodes, it only selects next action $a$ from current children. After a root-to-leaf search path is completed, $\theta$ and $V$ are updated through the search path from the leaf node as procedure *Back-Propagation* in Algorithm 1.

**Remark.** MuZero [4] selects nodes/actions in MCTS via (1) where the term $\sqrt{\sum_b N(s, b)}$ represents the total sampling time. In MALinZero, we utilize $\mathrm{trace}(V)$ to achieve the same effect. We use $\mathrm{trace}(V)$ rather than its square root due to the existence of $\sqrt{a^\top V^{-1} a}$ in LinUCT. It ensures that the scale of exploration term can keep stable with the increasing times of selection. Using the definition of $V$ and the fact that actions $A$ are $n$-hot vectors, it is easy to show that $\mathrm{trace}(V)$ increases linearly with $N$ and sampling time. For a single-agent problem, (10) indeed reduces to (1), recovering existing result as a special single-agent case.

With Dynamic Node Generation (DNG), we can sample and add new child nodes according to LinUCT. In other words, the $\kappa$ sampled child nodes are used to bootstrap a low-dimensional representation of the joint reward over the entire joint action space, thus enabling fast exploration and exploitation

in MALinZero. Let ground set $\mathcal{A}$ be the set of all $n$-hot vectors in $\mathbb{R}^{nd}$ where each vector $a \in \mathcal{A}$ satisfies: in each of the $n$ disjoint $k$-dimensional blocks, exactly one entry is 1 with others are 0. Let $\mathcal{S}$ be the set of selected actions and rewrite $V(\mathcal{S}) = \lambda I + \sum_{a \in \mathcal{S}} aa^\top$ using $\mathcal{S}$. We show that the objective function $\Psi(a)$ for action selection is sub-modular.

**Theorem 3.** *[Submodularity of $\Psi$] $\Psi$ is a non-negative monotonic submodular function over the ground set $\mathcal{A}$.*

Hence, to solve the optimization for action selection in (10), we have to maximize a submodular function, which is shown to be $NP$-hard [34, 35] by reduction from the classical Max-Coverage problem. Fortunately, there exists an $(1 - \frac{1}{e})$-approximation algorithm [36] to solve this optimization. Let $\Psi : 2^\mathcal{A} \to \mathbb{R}_{\geq 0}$ be a monotone submodular function. Fix a budget $T \in \mathbb{N}$ and let $\mathcal{A} = \bigsqcup_{i=1}^n B_i$ be partitioned into $n$ blocks (so that any feasible set contains at most one element from each $B_i$; i.e. an $n$-hot constraint).

**Theorem 4.** *[$(1 - \frac{1}{e})$-Approximation under Cardinality and $n$-Hot Constraints] There exists an [$(1 - \frac{1}{e})$-approximation algorithm for the optimization of action selection.*

*(a) Uniform-matroid (cardinality) case $|S| \leq T$. The standard greedy algorithm*

$$A_t = \arg\max_{a \in \mathcal{A} \setminus S_{t-1}} \left[ \Psi(S_{t-1} \cup \{a\}) - \Psi(S_{t-1}) \right], \quad S_t = S_{t-1} \cup \{A_t\},$$

*for $t = 1, \ldots, T$, returns $S_T$ satisfying $\Psi(S_T) \geq \left(1 - \frac{1}{e}\right) \Psi(S^\star)$, where $S^\star$ is an optimal subset of size at most $T$ [36].*

*(b) $n$-Hot (partition-matroid) case. One may apply the continuous-greedy algorithm to the multilinear relaxation $\max_{x \in P(\mathcal{M}),\ \mathbf{1}^\top x \leq T} \mathbb{E}[\Psi(R(x))]$, where $P(\mathcal{M})$ is the matroid polytope of the partition matroid and $R(x)$ denotes the standard randomised rounding. It produces a feasible set $\hat{S}$ with $\Psi(\hat{S}) \geq \left(1 - \frac{1}{e}\right) \Psi(S^\star)$ [37].*

Thus, under the stronger $n$-hot (partition-matroid) constraint, there exists an efficient algorithm to compute action selection in MALinZero with $(1 - \frac{1}{e})$-approximation.

---

**Algorithm 1** MALinZero

---

1: **procedure** DYNAMIC NODE GENERATION
2:     $a \leftarrow \arg\max_{a \in \mathcal{A}} a^\top \theta + c(s)P(s,a)\text{trace}(V)\sqrt{a^\top V^{-1} a}$
3:     **return** (s,a)
4: **end procedure**

1: **procedure** EXPANSION
2:     ▷ $M'$ is the number of nodes generated by sampling.
3:     **for** $i = 1, \ldots, M'$ **do**
4:         $a_i \leftarrow$ sample with $\beta$ and $P$ as Sampled MuZero[32]
5:         $T(s) \leftarrow T(s) \cup (s, a_i)$
6:     **end for**
7: **end procedure**

1: **procedure** SELECTION
2:     **if** number of child nodes < M **then**
3:         $(s, a) \leftarrow$ DYNAMIC NODE GENERATION
4:         $T(s) \leftarrow T(s) \cup (s, a)$
5:     **else**
6:         $a \leftarrow \arg\max_{a \in T(s)} a^\top \theta + c(s)P(s,a)\text{trace}(V)\sqrt{a^\top V^{-1} a}$
7:     **end if**
8:     **return** Index of $(s, a)$
9: **end procedure**

1: **procedure** BACK-PROPAGATION
2:     **for** $(s, a) \in$ path **do**
3:         Let $k, l$ be the depth of the current node $s$ and the leaf node.
            ▷ The weighting could be replaced with strongly-convex $\mu$-smooth function for better performance.
4:         **if** Observed reward $X_k \leq Q(s,a)$ **then**
5:             $w \leftarrow w_1$
6:         **else**
7:             $w \leftarrow w_2$
8:         **end if**
9:         Calculate the cumulative discounted reward $G(s) \leftarrow \sum_{\tau=0}^{l-1-k} \gamma^\tau X_{k+1+\tau} + \gamma^{l-k} v^l$
10:        $Q(s,a) \leftarrow \frac{N(s,a)Q(s,a)+G(s)}{N(s,a)+1}$
11:        $N(s,a) \leftarrow N(s,a) + 1$
12:        $V(s) \leftarrow V(s) + wa^\top a$
13:        $\theta(s) \leftarrow V(s)^{-1} X_k a$
14:     **end for**
15: **end procedure**

---

**Efficient Back-Propagation** The update of $\theta$ and $V$ involves large matrix manipulation, of which the time complexity is $\mathcal{O}(n^2 d^2)$ and the space complexity is $\mathcal{O}(n^2 d^2)$. To mitigate the computation complexity, we design an efficient back-propagation (as shown in Algorithm 1) to reduce both time and space complexity to $\mathcal{O}(nd)$ based on the Sherman-Morrison formula [38]. We consider the update of $A^T \hat{\theta}_t$ and $\sqrt{A^T V_t^{-1} A}$ in LinUCT. Using the definition of $V_t$ and $\hat{\theta}_t$, it is easy to show that these can be obtained by storing and recursively updating $\hat{\theta}_t$ and $V_t^{-1} A$:

$$V_{t+1}^{-1} A = V_t^{-1} A - \frac{V_t^{-1} A A^T V_t^T A_i}{1 + A^T V_t^T A} \text{ and } \hat{\theta}_{t+1} = V_t^{-1} M_t - \frac{V_t^{-1} A A^T V_t^{-1} M_t}{1 + A^T V_t^{-1} A},$$

where $A_i$ is the action corresponding to the $i$-th child node, $A$ is the action of nodes in the back-propagation path, and where $M_t = \sum_{s=1}^{t} w_s A_s X_s$ is an auxiliary variable.

**Theorem 5** (Complexity of the Back-Propagation to update $\hat{\theta}_t$ and $V_t^{-1} A$). *The proposed method computes the same LinUCT, but reduces the computation complexity from $\mathcal{O}(n^2 d^2)$ to $\mathcal{O}(nd)$.*

## 4 Experiments

We evaluate MALinZero on three reinforcement learning benchmarks: MatGame, StarCraft Multi-Agent Challenge (SMAC)[22] and SMACv2 [23]. MatGam is a stateless-matrix game that generalizes the classic normal-form setting to $n$ agents. At every step, all agents select an action from the same discrete set; the environment then looks up the joint action in a predefined payoff (with or without noise) tensor and returns the corresponding shared reward, which is used to evaluate algorithms' performance. MALinZero is compared with both model-based and model-free baseline models on these environments. The model-based algorithms are MAZero [10], MAZero without prior information (MAZero-NP) and MuZero implemented for multi-agent tasks (MA-AlphaZero). We also choose two mainstream model-free MARL algorithms: MAPPO [39] and QMIX [14].

**Model architecture** MALinZero consists of 6 neural networks to be learned during the training and the parameter $\theta$ is to be estimated from initialization for a single MCTS process. Specifically, with network parameter $\phi$, there are 6 key functions: the representation function $s_{t,0}^i = h_\phi(o_{<t}^i)$ that maps the current individual observation history into the latent space, the communication function $e_{t,k}^1, \ldots, e_{t,k}^n = e_\phi(s_{t,k}^1, \ldots, s_{t,k}^n, a_{t,k}^1, \ldots, a_{t,k}^n)$ that generates cooperative information for each agent via the attention mechanism, the dynamic function $s_{t,k+1}^i = g_\phi(s_{t,k}^i, a_{t+k}^i, e_{t,k}^i)$ that plays the role of transition function, the reward function $r_{t,k} = r_\phi(s_{t,k}^1, \ldots, s_{t,k}^n, a_{t,k}^1, \ldots, a_{t,k}^n)$ and the value function $v_{t,k} = v_\phi(s_{t,k}^1, \ldots, s_{t,k}^n)$ that predicts the reward and value respectively, and the policy function $p_{t,k}^i = p_\phi(s_{t,k}^i)$ that predicts the policy distribution for the given state. The subscript $k$ denotes the index of unrolling steps within one simulation from the root node in MCTS. The update of estimated $\theta$ takes place in the Back-propagation stage and the detailed process is analyzed above. For all these modules except for the communication function $e_\phi$, the neural networks are implemented by Multi-Layer Perception (MLP) networks and a Rectified Linear Unit (ReLU) activation and Layer Normalization (LN) follows each linear layer in MLP networks. Agents process local dynamics and make predictions with the encoded information.

**Experiment setting** All experiments are conducted using NVIDIA RTX A6000 GPUs and NVIDIA A100 GPUs. For MatGame environments, the number of sampled actions for each node in MCTS is 3 and the number of MCTS simulations is 50. For both SMAC and SMACv2 benchmarks, we set them as 7 and 100, respectively. We build our training pipeline similar to EfficientZero [40] which synchronizes parallel stages of data collection, reanalysis, and training.

**Performance Evaluation** MALinZero outperforms all baselines in 8 MatGame environments. As shown in Table 1, the performance improvements are achieved in even simple MatGames (a few percent for 2 agents with 3 actions each, thus a space of only 9 joint actions) and increases for more complex MatGames (such as up to 11% for 8 agents each with 10 actions, thus a space of $8^{10}$ joint actions). This makes sense since the benefit of MALinZero comes from representing high-dimensional joint action space into lower-dimensional ones. Interestingly, the improvements are higher in MatGames with non-linear reward structures. This is because MALinZero is able to model

| Agent | Action | Type | Steps | MAZero | MAZero-NP | MA-AlphaZero | MAPPO | QMIX | MALinZero(Ours) |
|---|---|---|---|---|---|---|---|---|---|
| 2 | 3 | Linear | 500 | $51.9 \pm 2.3$ | $49.7 \pm 3.9$ | $50.8 \pm 3.2$ | $50.2 \pm 2.9$ | $50.4 \pm 3.5$ | $\mathbf{53.1 \pm 0.9}$ |
| 2 | 3 | Linear | 1000 | $57.8 \pm 2.4$ | $53.1 \pm 3.3$ | $55.2 \pm 2.7$ | $56.4 \pm 3.1$ | $54.3 \pm 3.17$ | $\mathbf{59.9 \pm 0.2}$ |
| 2 | 3 | Non-Linear | 500 | $49.1 \pm 15.3$ | $48.9 \pm 17.2$ | $49.0 \pm 16.4$ | $49.1 \pm 19.1$ | $48.7 \pm 18.6$ | $\mathbf{49.2 \pm 8.6}$ |
| 2 | 3 | Non-Linear | 1000 | $47.6 \pm 14.7$ | $49.3 \pm 14.3$ | $49.2 \pm 12.9$ | $49.5 \pm 18.1$ | $49.1 \pm 17.7$ | $\mathbf{49.6 \pm 15.5}$ |
| 4 | 5 | Linear | 1000 | $175.2 \pm 4.4$ | $171.7 \pm 5.6$ | $172.7 \pm 4.1$ | $173.1 \pm 5.4$ | $171.8 \pm 4.9$ | $\mathbf{184.3 \pm 3.2}$ |
| 4 | 5 | Linear | 2000 | $191.7 \pm 2.3$ | $190.1 \pm 1.2$ | $190.4 \pm 1.9$ | $189.8 \pm 2.1$ | $190.2 \pm 1.8$ | $\mathbf{197.4 \pm 2.1}$ |
| 4 | 5 | Non-Linear | 1000 | $179.4 \pm 11.7$ | $173.2 \pm 10.0$ | $174.5 \pm 9.3$ | $173.1 \pm 8.0$ | $174.7 \pm 9.4$ | $\mathbf{182.4 \pm 11.7}$ |
| 4 | 5 | Non-Linear | 2000 | $195.4 \pm 20.0$ | $192.4 \pm 12.8$ | $192.7 \pm 11.4$ | $191.9 \pm 12.5$ | $190.3 \pm 10.7$ | $\mathbf{197.8 \pm 21.1}$ |
| 6 | 8 | Linear | 1000 | $393.7 \pm 9.9$ | $387.2 \pm 10.1$ | $389.3 \pm 8.4$ | $390.6 \pm 9.2$ | $386.1 \pm 10.4$ | $\mathbf{396.6 \pm 8.4}$ |
| 6 | 8 | Linear | 2000 | $434.2 \pm 7.2$ | $427.3 \pm 9.3$ | $432.6 \pm 9.5$ | $431.8 \pm 8.4$ | $430.1 \pm 9.5$ | $\mathbf{439.8 \pm 6.8}$ |
| 6 | 8 | Non-Linear | 1000 | $399.8 \pm 13.7$ | $391.3 \pm 10.3$ | $393.1 \pm 12.1$ | $388.8 \pm 13.1$ | $390.5 \pm 12.2$ | $\mathbf{410.6 \pm 8.9}$ |
| 6 | 8 | Non-Linear | 2000 | $443.9 \pm 12.1$ | $429.1 \pm 9.3$ | $427.1 \pm 8.6$ | $430.1 \pm 8.5$ | $431.7 \pm 7.6$ | $\mathbf{451.1 \pm 12.8}$ |
| 8 | 10 | Linear | 1000 | $618.8 \pm 16.9$ | $608.8 \pm 17.6$ | $613.1 \pm 13.1$ | $617.1 \pm 11.1$ | $612.7 \pm 15.4$ | $\mathbf{637.1 \pm 15.8}$ |
| 8 | 10 | Linear | 2000 | $692.7 \pm 14.5$ | $671.5 \pm 13.9$ | $654.3 \pm 14.5$ | $681.8 \pm 12.5$ | $679.4 \pm 12.7$ | $\mathbf{705.2 \pm 15.7}$ |
| 8 | 10 | Non-Linear | 1000 | $615.2 \pm 18.7$ | $536.6 \pm 24.1$ | $573.2 \pm 22.7$ | $561.4 \pm 20.9$ | $558.7 \pm 19.1$ | $\mathbf{630.1 \pm 16.3}$ |
| 8 | 10 | Non-Linear | 2000 | $672.3 \pm 16.1$ | $587.2 \pm 18.4$ | $633.2 \pm 15.6$ | $657.1 \pm 17.3$ | $648.2 \pm 18.75$ | $\mathbf{693.4 \pm 15.6}$ |

Table 1: Evaluation in MatGame with different numbers of agents and actions. We consider both linear and non-linear reward structures. MALinZero is shown to outperform both MCTS and MARL baselines, especially in more complex MatGames with larger action spaces and with less numbers of steps. Interestingly, the improvement is higher for non-linear reward structures (up to %11), as baselines may stuck in local optima. Detailed MatGame settings can be found in Appendix D.

the entire joint action space – despite in a lower dimensional space, while baselines may get stuck in local optima. MALinZero is also able to achieve the rewards much faster than baselines. Running the LinUCB algorithm will incur minor additional cost. However, the computation leverages a linear structure with sampling $\mathcal{O}(dn)$ actions rather than the standard $\mathcal{O}(d^n)$. Our evaluation shows that the computational cost is comparable to that of the MAZero [10] method.

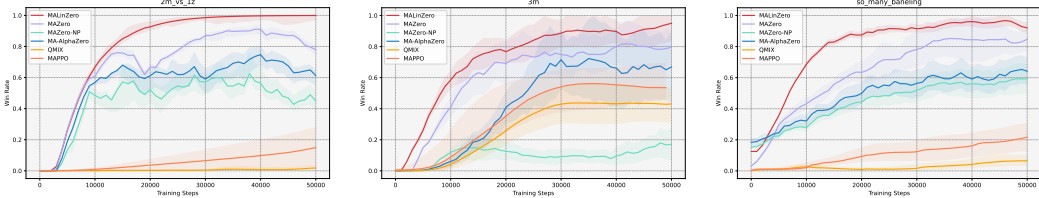

Figure 1: Evaluations on 3 SMAC tasks/maps. Y-axis denotes the win rate and X-axis denotes training steps. Each algorithm is executed with 3 random seeds. MALinZero achieves over 95% winning rate on all 3 maps, outperforming all baselines and also gets high winning rate much faster.

Figure 1 shows performance measured by win rate on three different SMAC maps. MALinZero beats all five MCTS and MARL baselines, in both higher winning rate (over 95% across all maps) and faster convergence speed. Comparing with the closest baseline MAZero, our MALinZero reaches the same winning rate with 50% to 70% less steps/samples, implying 2-3× speedup. The results demonstrate LinUCT's ability to represent complex multi-agent decision-making problems in low-dimensional latent space. This efficient representation supports fast MCTS by exploring and exploiting the global reward structure of the joint action space (in an approximated low-dimensional fashion), rather than getting trapped in local optima as in the baselines. This is validated by comparison with MCTS baselines with pUCT applied to MAZero, MAZero-NP, and MA-AlphaZero.

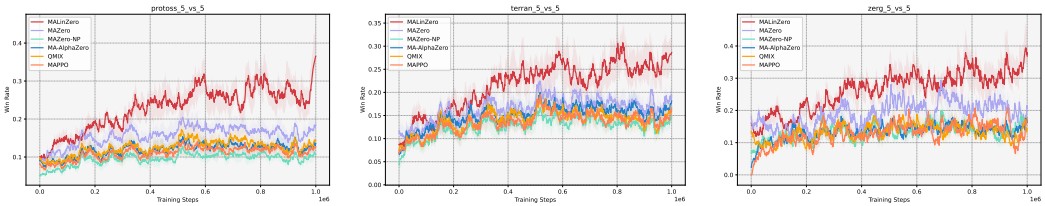

Figure 2: Comparisons on 3 SMACv2 tasks/maps.Y-axis denotes the win rate and X-axis denotes the training steps. MALinZero nearly doubles the winning rate on these challenging maps in SMACv2 and consistently outperforms all baselines. Each algorithm is executed with 3 random seeds.

Different from SMAC, SMACv2 significantly increases difficulty by adding larger heterogeneous unit teams, more varied map layouts, and stochastic enemy formations, which all demand advanced coordination and generalization by the learning algorithms. Figure 2 shows the training curves of our proposed MALinZero and baseline algorithms on SMACv2, including 3 widely-used maps. Compared with all baselines, MALinZero doubles the winning rate on protoss_5_vs_5 and zerg_5_vs_5, and nearly doubles it on terran_5_vs_5. Our MALinZero shows very robust performance across different scenarios, which comes from the parameterization of LinUCT, allowing MALinZero to conduct more adaptive and efficient modeling of heterogeneous unit teams.

**Ablation Study**   We intend to validate the necessity and effectiveness of DNG and the general function $f$ applied in LinUCT. To accomplish this, we compare the proposed MALinZero under two MatGame environments: (1) Medium difficulty scenario containing 4 agents and each with 5 actions; (2) Hard difficulty scenario containing 8 agents and each with 10 actions.

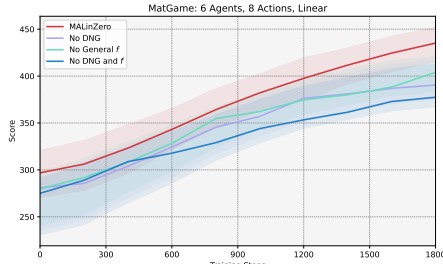 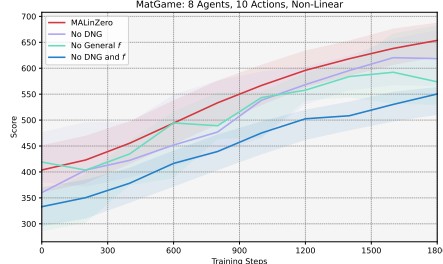

Figure 3: Ablation study of MALinZero by removing various design components, such as DNG and the introduction of general convex loss $f$ in the contextual bandit problem.

In Figure 3, we evaluate the impact of removing the DNG component, the use of the general convex loss $f$ (to place more importance on better actions), and both simultaneously. It is shown that these components are critical for the superior performance of MALinZero. In particular, without DNG, it is hard for MALinZero to model and explore the joint action space, thus the performance becomes limited. The observed performance degradation when using a Euclidean distance rather than general convex loss $f$ validates our design principle that by placing more importance on the better actions can boost maximal action selection in this low-dimension representation.

## 5   Conclusions

We propose MALinZero, which leverages low-dimensional representational structures to enable efficient MCTS in complex multi-agent planning. MALinZero can be viewed as projecting the joint-action returns into the low-dimensional space representable using a contextual linear bandit problem formulation, with a convex and $\mu$-smooth loss to place more importance on better actions. We employ an $(1 - \frac{1}{e})$-approximation algorithm for the joint action selection by maximizing a submodular objective. MALinZero demonstrates state-of-the-art performance on multi-agent benchmarks such as MatGame, SMAC, and SMACv2, outperforming MARL and MCTS baselines.

**Limitations:** MALinZero leverages a contextual linear bandit formulation in the low-dimensional space. The use of non-linear formulations that may also allow efficient MCTS could further improve the performance. Developing fully decomposable representations also remains an open problem.

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

# A  Proof of Theorems

## A.1  Proof of Theorem 1

We will show the proof for the regret bound of our proposed LinUCT.

**Setup.** Let $\theta^* \in \mathbb{R}^{nd}$, $\|\theta^*\|_2 \leq S$. At each round $t$ we observe $A_t \in \mathbb{R}^{nd}$, $\|A_t\|_2 \leq \sqrt{n} = L$, and receive $X_t = \langle \theta^*, A_t \rangle + \eta_t$, where $\eta_t$ is conditionally 1-sub-Gaussian. Suppose $f : \mathbb{R} \to \mathbb{R}$ is strongly-convex and $\mu$-smooth, we have $f''(z) \in [\varepsilon, \mu], \forall z \in \mathbb{R}$ for some positive $\varepsilon$. The solution to (2) is obtained by differentiation and yields $\hat{\theta}_t = V_t^{-1} \sum_{s=1}^t w_s A_s X_s$ where we use $w_t = f''(\xi_t)$ with $\xi_t \in (0, X_t - \langle \theta_{t-1}, A_t \rangle)$ and thus have $\varepsilon \leq w_t \leq \mu$ for any $t$ and $\xi_t$.

We consider an ellipsoid confidence set centered around the optimal estimator $\hat{\theta}_{t-1}$, i.e., $\mathcal{C}_t = \left\{ \theta \in \mathbb{R}^{nd} : \|\theta - \hat{\theta}_{t-1}\|_{V_{t-1}} \right\} \leq \beta_t$, for any increasing sequence of $\beta_t$ with $\beta_1 \geq 1$. Note that as $t$ grows, this ellipse $\mathcal{C}_t$ is shrinking as $V_t$ has increasing eigenvalues and if $\beta_t$ does not grow too fast. We show that the problem of selecting optimal action $A_t \in \mathcal{A}$ by solving $\max_{A_t \in \mathcal{A}, \theta \in \mathcal{C}_t} \langle \theta, a \rangle$ in this contextual linear bandit problem is equivalent to $A_t = \arg\max_a \left\langle \hat{\theta}_{t-1}, a \right\rangle + \beta_{t-1} \|a\|_{V_{t-1}^{-1}}$, which is referred to as our LinUCT rule for action selection. We consider the realized regret defined by $\widehat{R}_T = \sum_{t=1}^T (X_t^* - X_t) = \sum_{t=1}^T \left( \langle \theta^*, A_t^* \rangle - \langle \theta^*, A_t \rangle \right) + \sum_{t=1}^T (\eta_t^* - \eta_t)$.

**Lemma 1** (Confidence Ellipsoid). *With probability at least $1 - \delta$, for all $t \leq T$,*

$$\|\theta_t - \theta^*\|_{V_t} \leq \beta_t. \tag{11}$$

*Proof.* Observe

$$\theta_t - \theta^* = V_t^{-1} \left( \sum_{s=1}^t w_s A_s X_s - V_t \theta^* \right) = V_t^{-1} \left( \sum_{s=1}^t w_s A_s \eta_s - \lambda \theta^* \right). \tag{12}$$

Set $Y_t = \sum_{s=1}^t w_s A_s \eta_s$ and $b = \lambda \theta^*$. Then we have

$$\|\theta_t - \theta^*\|_{V_t}^2 = (b - Y_t)^\top V_t^{-1} (b - Y_t) = Y_t^\top V_t^{-1} Y_t - 2 b^\top V_t^{-1} Y_t + b^\top V_t^{-1} b. \tag{13}$$

Since $V_t \succeq \lambda I$, $V_t^{-1} \preceq \frac{1}{\lambda} I$ and $\|\theta^*\| \leq S$, we can get

$$b^\top V_t^{-1} b = \lambda^2 \theta^{*\top} V_t^{-1} \theta^* \leq \lambda^2 \frac{S^2}{\lambda} = \lambda S^2. \tag{14}$$

According to Cauchy–Schwarz inequality and $V_t^{-1} \preceq \frac{1}{\lambda} I$,

$$|b^\top V_t^{-1} Y_t| \leq \|b\|_2 \|V_t^{-1} Y_t\|_2 \leq \lambda S \sqrt{\frac{1}{\lambda} Y_t^\top V_t^{-1} Y_t} = \sqrt{\lambda} S \sqrt{Y_t^\top V_t^{-1} Y_t}. \tag{15}$$

Hence

$$-2 b^\top V_t^{-1} Y_t \leq 2 \sqrt{\lambda} S \sqrt{Y_t^\top V_t^{-1} Y_t}. \tag{16}$$

To bound $Y_t^\top V_t^{-1} Y_t$, note $Y_t = \sum_{s=1}^t w_s A_s \eta_s$ is a martingale sum. According to Lemma 2, the self-normalized tail bound yields

$$Y_t^\top V_t^{-1} Y_t \leq 2\mu \ln \frac{\det(V_t)^{1/2}}{\det(\lambda I)^{1/2} \delta}. \tag{17}$$

Combining these three yields

$$\|\theta_t - \theta^*\|_{V_t} \leq \sqrt{Y_t^\top V_t^{-1} Y_t} + \sqrt{\lambda} S \leq \beta_t, \tag{18}$$

and a union bound over $t = 1, \ldots, n$ gives the result. $\qquad\square$

**Lemma 2** (Self-Normalized Martingale Tail). *Let $\{\eta_t\}_{t=1}^T$ be a sequence of conditionally 1-sub-Gaussian noises, and let $A_t \in \mathbb{R}^n d$ and $w_t \in [\varepsilon, \mu]$ be $\mathcal{F}_{t-1}$-measurable. Define*

$$Y_t \;=\; \sum_{s=1}^t w_s\, A_s\, \eta_s, \; V_t \;=\; \sum_{s=1}^t w_s\, A_s A_s^\top + \lambda I. \tag{19}$$

*Then for any $\delta \in (0,1)$, with probability at least $1 - \delta$ simultaneously for all $t \le T$,*

$$Y_t^\top V_t^{-1} Y_t \;\le\; 2\mu\, \ln\!\Big(\frac{\det(V_t)^{1/2}}{\det(\lambda I)^{1/2}\,\delta}\Big). \tag{20}$$

*Proof.* First, we define

$$M_t(x) \;=\; \exp\!\Big(x^\top Y_t \;-\; \tfrac{\mu}{2}\, x^\top V_t\, x\Big) \tag{21}$$

for each fixed $x \in \mathbb{R}^n d$. Since $\eta_t$ is conditionally 1-sub-Gaussian and $w_t \le \mu$, we have for any $\mathcal{F}_{t-1}$-measurable $u$

$$\mathbb{E}\big[e^{u\,\eta_t} \mid \mathcal{F}_{t-1}\big] \;\le\; \exp\big(\tfrac{1}{2}u^2\big) \implies \mathbb{E}\big[e^{w_t\, A_t^\top x\, \eta_t} \mid \mathcal{F}_{t-1}\big]$$
$$\le\; \exp\big(\tfrac{1}{2}w_t^2(A_t^\top x)^2\big) \;\le\; \exp\big(\tfrac{\mu}{2}\, x^\top (w_t A_t A_t^\top)\, x\big). \tag{22}$$

Therefore

$$\mathbb{E}\big[M_t(x) \mid \mathcal{F}_{t-1}\big] = M_{t-1}(x)\, \mathbb{E}\Big[e^{x^\top (w_t A_t \eta_t) - \frac{\mu}{2} x^\top (w_t A_t A_t^\top) x} \,\Big|\, \mathcal{F}_{t-1}\Big] \;\le\; M_{t-1}(x). \tag{23}$$

Hence each $M_t(x)$ is a nonnegative supermartingale with $M_0(x) = 1$.

Then we lift the pointwise supermartingale bound to a uniform one by integrating $M_t(x)$ against the Gaussian prior over $x$. Let $h$ be the density of $\mathcal{N}(0, \lambda^{-1}I)$. Define the mixture

$$\overline{M}_t \;=\; \int_{\mathbb{R}^n d} M_t(x)\, h(x)\, dx. \tag{24}$$

By Fubini and the supermartingale property,

$$\mathbb{E}[\overline{M}_t \mid \mathcal{F}_{t-1}] = \int \mathbb{E}[M_t(x) \mid \mathcal{F}_{t-1}]\, h(x)\, dx \;\le\; \int M_{t-1}(x)\, h(x)\, dx = \overline{M}_{t-1}, \tag{25}$$

so $\overline{M}_t$ is also a nonnegative supermartingale with $\overline{M}_0 = 1$. A Gaussian integral gives

$$\overline{M}_t = \frac{1}{(2\pi)^{nd/2} \det(\lambda^{-1}I)^{1/2}} \int \exp\!\Big(x^\top Y_t - \tfrac{1}{2}x^\top(\lambda I + \mu V_t)x\Big)\, dx$$
$$= \Big(\tfrac{\det(\lambda I)}{\det(\lambda I + \mu V_t)}\Big)^{1/2} \exp\!\Big(\tfrac{1}{2}Y_t^\top(\lambda I + \mu V_t)^{-1}Y_t\Big). \tag{26}$$

Since $V_t \succeq \lambda I$, one checks $(\lambda I + \mu V_t)^{-1} \succeq \tfrac{1}{\mu}V_t^{-1}$, and $\det(\lambda I + \mu V_t) \le \mu^{nd}\det(V_t)$. Thus

$$\overline{M}_t \;\ge\; \mu^{-nd/2}\Big(\tfrac{\det(\lambda I)}{\det(V_t)}\Big)^{1/2} \exp\!\Big(\tfrac{1}{2\mu}Y_t^\top V_t^{-1}Y_t\Big). \tag{27}$$

By Ville's maximal inequality for nonnegative supermartingales,

$$\Pr\!\Big(\exists\, t \le T : \overline{M}_t \ge \tfrac{1}{\delta}\Big) \;\le\; \delta\, \overline{M}_0 = \delta. \tag{28}$$

On the complementary event, for all $t \le T$,

$$\overline{M}_t < \tfrac{1}{\delta} \implies \tfrac{1}{2\mu}Y_t^\top V_t^{-1}Y_t \;\le\; \tfrac{nd}{2}\ln\mu \;+\; \tfrac{1}{2}\ln\frac{\det(V_t)}{\det(\lambda I)} \;+\; \ln\frac{1}{\delta}. \tag{29}$$

Absorbing the constant $\tfrac{nd}{2}\ln\mu$ into $\ln(1/\delta)$ yields

$$Y_t^\top V_t^{-1}Y_t \;\le\; 2\mu\, \ln\!\Big(\tfrac{\det(V_t)^{1/2}}{\det(\lambda I)^{1/2}\delta}\Big), \tag{30}$$

as claimed. $\qquad\square$

**Lemma 3** (Elliptical Potential). *Let $V_0 = \lambda I$ and for $t = 1, 2, \ldots, T$ define*

$$V_t = V_{t-1} + w_t A_t A_t^\top, \tag{31}$$

*where $A_t \in \mathbb{R}^{nd}$ satisfies $\|A_t\|_2 \le \sqrt{n} = L$ and $w_t \in [\varepsilon, \mu]$ with $\varepsilon \ge 0$. Then*

$$\sum_{t=1}^{T} \min\left\{1, \ w_t \|A_t\|_{V_{t-1}^{-1}}^2\right\} \ \le \ 2 \ln\frac{\det(V_T)}{\det(V_0)} \ \le \ 2\,nd \ln\left(1 + \tfrac{\mu T}{d\lambda}\right). \tag{32}$$

*Proof.* First, for any $z \ge 0$ we have $z \wedge 1 \le 2\ln(1 + z)$. Hence

$$\sum_{t=1}^{T} \min\{1, w_t\|A_t\|_{V_{t-1}^{-1}}^2\} \ \le \ 2\sum_{t=1}^{T} \ln\left(1 + w_t \|A_t\|_{V_{t-1}^{-1}}^2\right). \tag{33}$$

Next, by the matrix determinant lemma,

$$\det(V_t) = \det(V_{t-1})\det\left(I + w_t V_{t-1}^{-1/2} A_t A_t^\top V_{t-1}^{-1/2}\right) = \det(V_{t-1})\left(1 + w_t \|A_t\|_{V_{t-1}^{-1}}^2\right). \tag{34}$$

Telescoping the product for $t = 1, \ldots, T$ gives

$$\prod_{t=1}^{T}\left(1 + w_t \|A_t\|_{V_{t-1}^{-1}}^2\right) = \frac{\det(V_T)}{\det(V_0)}, \tag{35}$$

and taking logarithms,

$$\sum_{t=1}^{T} \ln\left(1 + w_t \|A_t\|_{V_{t-1}^{-1}}^2\right) = \ln\frac{\det(V_T)}{\det(V_0)}. \tag{36}$$

Combining with the earlier bound yields the first inequality. Finally, since $w_t \le \mu$ and $\|A_t\| \le L$, we have

$$V_T = \lambda I + \sum_{t=1}^{T} w_t A_t A_t^\top \ \preceq \ \lambda I + \mu L^2 T\, I, \tag{37}$$

so

$$\ln\frac{\det(V_T)}{\det(\lambda I)} \ \le \ nd \ln\left(\frac{nd\lambda + \mu L^2 T}{nd\lambda}\right) = nd \ln\left(1 + \tfrac{\mu T}{d\lambda}\right), \tag{38}$$

giving the second inequality. $\qquad\square$

Here we reclaim Theorem 1

**Theorem.** *1 [Regret Bound of LinUCT] With probability $1 - \delta$, the regret of LinUCT satisfies*

$$\hat{R}_t \le \sqrt{8\mu t \beta_t \ln\left(\frac{\det(V_t)}{\det(\lambda I)}\right)} \le \sqrt{8\mu n d t \beta_t \ln\left(\frac{nd\lambda + \mu n t}{nd\lambda}\right)}. \tag{39}$$

*Proof.* Let

$$S_t = \sum_{s=1}^{t} w_s A_s \eta_s, \qquad V_t = \lambda I + \sum_{s=1}^{t} w_s A_s A_s^\top, \tag{40}$$

and define

$$\beta_t = \sqrt{2\mu \ln\frac{\det\left(V_t\right)^{1/2}}{\lambda^{nd/2}\,\delta}} \ + \ \sqrt{\lambda}\,\|\theta^*\|_2. \tag{41}$$

By Lemma 2, with probability at least $1 - \delta$ simultaneously for all $t$,

$$\|S_t\|_{V_t^{-1}} \ \le \ \sqrt{2\mu \ln\frac{\det\left(V_t\right)^{1/2}}{\lambda^{nd/2}\,\delta}} \ = \ \beta_t - \sqrt{\lambda}\,\|\theta^*\|_2. \tag{42}$$

On this event, Lemma 1 shows

$$\|\hat{\theta}_t - \theta^*\|_{V_t(\lambda)} \leq \|S_t\|_{V_t(\lambda)^{-1}} + \sqrt{\lambda}\,\|\theta^*\|_2 \leq \beta_t. \tag{43}$$

Next let $\Delta_t = \eta_t^* - \eta_t$. Since each $\eta_t, \eta_t^*$ is 1-sub-Gaussian and independent, we can get

$$\mathbb{E}\big[e^{\lambda \Delta_t} \mid \mathcal{F}_{t-1}\big] = \mathbb{E}\big[e^{\lambda \eta_t^*}\big]\,\mathbb{E}\big[e^{-\lambda \eta_t}\big] \leq \exp\!\Big(\tfrac{\lambda^2}{2}\Big)\exp\!\Big(\tfrac{\lambda^2}{2}\Big) = \exp\!\Big(\lambda^2\Big). \tag{44}$$

Thus $\Delta_t$ is conditionally $\sqrt{2}$-*sub-Gaussian*:

$$\mathbb{E}\big[e^{\lambda \Delta_t} \mid \mathcal{F}_{t-1}\big] \leq \exp\!\Big(\tfrac{(\sqrt{2}\lambda)^2}{2}\Big). \tag{45}$$

By Hoeffding's inequality,

$$\Pr\!\Big(\sum_{s=1}^{t} \Delta_t > u\Big) \leq \exp\!\Big(-\tfrac{u^2}{4T}\Big). \tag{46}$$

Choose $u = 2\sqrt{T \ln \frac{1}{\delta}}$. Then

$$\Pr\!\Big(\sum_{s=1}^{t} \Delta_t > 2\sqrt{T \ln \tfrac{1}{\delta}}\Big) \leq \exp\!\Big(-\tfrac{4T \ln(1/\delta)}{4T}\Big) = \delta. \tag{47}$$

Because $A_t$ is chosen by $A_t = \arg\max_a \langle \hat{\theta}_{t-1}, a\rangle + \beta_{t-1}\|a\|_{V_{t-1}^{-1}}$, while $A_t^* = \arg\max_a \langle \theta^*, a\rangle$, we first compare the optimistic upper–confidence values:

$$\langle \hat{\theta}_{t-1}, A_t\rangle + \beta_{t-1}\|A_t\|_{V_{t-1}^{-1}} \geq \langle \hat{\theta}_{t-1}, A_t^*\rangle + \beta_{t-1}\|A_t^*\|_{V_{t-1}^{-1}}. \tag{48}$$

Whenever the confidence event (43) holds for any $a$,

$$|\langle (\theta^* - \hat{\theta}_{t-1}), a\rangle| \leq \|\theta^* - \hat{\theta}_{t-1}\|_{V_{t-1}}\,\|a\|_{V_{t-1}^{-1}} \leq \beta_{t-1}\|a\|_{V_{t-1}^{-1}}, \tag{49}$$

Applying this with $a = A_t^*$ and then with $a = A_t$ gives

$$\langle \theta^*, A_t^*\rangle \leq \langle \hat{\theta}_{t-1}, A_t^*\rangle + \beta_{t-1}\|A_t^*\|_{V_{t-1}^{-1}}, \qquad \langle \theta^*, A_t\rangle \geq \langle \hat{\theta}_{t-1}, A_t\rangle - \beta_{t-1}\|A_t\|_{V_{t-1}^{-1}}. \tag{50}$$

Subtracting the second inequality from the first and using the choice of $A_t$,

$$\langle \theta^*, A_t^*\rangle - \langle \theta^*, A_t\rangle \leq \beta_{t-1}\,\|A_t\|_{V_{t-1}^{-1}}. \tag{51}$$

Then we can get

$$X_t^* - X_t = \big[\langle \theta^*, A_t^*\rangle - \langle \theta^*, A_t\rangle\big] + \Delta_t \leq \beta_{t-1}\|A_t\|_{V_{t-1}^{-1}} + \Delta_t. \tag{52}$$

The single-step regret is

$$r_t := X_t^* - X_t = \langle \theta^*, A_t^*\rangle - \langle \theta^*, A_t\rangle + \Delta_t \leq \beta_{t-1}\|A_t\|_{V_{t-1}^{-1}} + \Delta_t. \tag{53}$$

Sum the single-step regret from $t = 1$ to $T$:

$$\hat{R}_T := \sum_{t=1}^{T} r_t \leq \sum_{t=1}^{T} \beta_{t-1}\,\|A_t\|_{V_{t-1}^{-1}} + 2\sqrt{T \ln \tfrac{1}{\delta}}. \tag{54}$$

Inequality (53) is the starting point for the final bounding of the main term

$$\sum_{t=1}^{T} \beta_{t-1}\|A_t\|_{V_{t-1}^{-1}} \tag{55}$$

via Cauchy–Schwarz together with the weighted elliptical potential lemma.

By Cauchy–Schwarz and Lemma 3,

$$\sum_{t=1}^{T} \beta_{t-1} \|A_t\|_{V_{t-1}^{-1}} \le \sqrt{\sum_{t=1}^{T} \beta_{t-1}^2} \sqrt{\sum_{t=1}^{T} \|A_t\|_{V_{t-1}^{-1}}^2} \le \sqrt{T} \, \beta_T \sqrt{2 \ln \frac{\det(V_T)}{\lambda^{nd}}} \, . \tag{56}$$

Combined with (54),

$$\hat{R}_T \le \sqrt{2T} \, \beta_T \sqrt{\ln \frac{\det(V_T)}{\lambda^{nd}}} + 2\sqrt{T \ln \tfrac{1}{\delta}}. \tag{57}$$

According to the definition of $\beta_T$, we have

$$\beta_T \le \sqrt{2\mu \ln \frac{\det(V_T)}{\lambda^{nd}}} \implies \ln \frac{\det(V_T)}{\lambda^{nd}} \le \frac{\beta_T^2}{2\mu}. \tag{58}$$

Therefore the first term in (57) satisfies

$$\sqrt{2T} \, \beta_T \sqrt{\ln \frac{\det(V_T)}{\lambda^{nd}}} \le \sqrt{2T} \, \beta_T \sqrt{\frac{\beta_T^2}{2\mu}} = \sqrt{\frac{T}{\mu}} \, \beta_T^2. \tag{59}$$

Moreover,

$$2\sqrt{T \ln \tfrac{1}{\delta}} \le 2\sqrt{T \frac{\beta_T^2}{2\mu}} = \sqrt{\frac{2T}{\mu}} \, \beta_T \le \sqrt{\frac{T}{\mu}} \, \beta_T^2. \tag{60}$$

Hence

$$\hat{R}_T \le 2\sqrt{\frac{T}{\mu}} \, \beta_T^2 = 2\sqrt{\frac{T}{\mu}} \left( \sqrt{2\mu \ln \frac{\det(V_T)^{1/2}}{\lambda^{nd/2}\delta}} + \sqrt{\lambda} \, \|\theta^*\|_2 \right)^2, \tag{61}$$

With $w_t \le \mu$ and $\|A_t\|_2 \le \sqrt{n} = L$, according to Lemma 3 we have

$$V_T = \lambda I + \sum_{s=1}^{T} w_s A_s A_s^\top \preceq nd\lambda I + \mu L^2 T \, I, \tag{62}$$

and

$$\det(V_T) \le \left( \frac{nd\lambda + \mu L^2 T}{nd} \right)^{nd}, \qquad \frac{\det(V_T)}{\lambda^{nd}} \le \left( \frac{nd\lambda + \mu L^2 T}{nd\lambda} \right)^{nd}. \tag{63}$$

Thus, with probability at least $1 - \delta$,

$$\hat{R}_t \le \sqrt{8\mu \, t \, \beta_t \left( \frac{nd\lambda + \mu L^2 t}{nd\lambda} \right)^{nd}} = \sqrt{8\mu nd \, t \, \beta_t \, \ln \left( \frac{nd\lambda + \mu nt}{nd\lambda} \right)}. \tag{64}$$

$\square$

## A.2 Proof of Theorem 3

**Theorem.** *3 [Submodularity of $\Psi$] $\Psi$ is a non-negative monotonic submodular function over the ground set $\mathcal{A}$.*

*Proof.* Throughout, $\|x\|_M := \sqrt{x^\top M x}$ for $M \succ 0$.

**(i) Non-negativity.** Both summands in $\Psi(S)$ are non-negative, hence $\Psi(S) \ge 0$ for all $S \subseteq \mathcal{A}$.

**(ii) Monotonicity.**   Fix $S \subseteq \mathcal{A}$ and $a \notin S$. Write

$$\Delta(a \mid S) \; = \; \Psi(S \cup \{a\}) - \Psi(S) \; = \; a^\top \theta + \underbrace{\|a\|_{V(S \cup \{a\})^{-1}}}_{\text{new radius}} - \sum_{v \in S} \Big( \|v\|_{V(S)^{-1}} - \|v\|_{V(S \cup \{a\})^{-1}} \Big).$$

(65)

Apply Lemma 4 with $V := V(S)$ and $u := a$:

$$\sum_{v \in S} \Big( \|v\|_{V(S)^{-1}} - \|v\|_{V(S \cup \{a\})^{-1}} \Big) \; \leq \; \|a\|_{V(S)^{-1}} \; \leq \; \|a\|_{V(S \cup \{a\})^{-1}}.$$

(66)

Substituting inequality (66) in (65) gives $\Delta(a \mid S) \geq a^\top \theta \geq 0$; therefore $\Psi$ is monotone.

**(iii) Submodularity (diminishing returns).**   Let $S \subseteq T \subseteq \mathcal{A}$ and let $a \notin T$. Set $U := T \setminus S$. For any finite $R \subseteq \mathcal{A}$ define

$$L(R) \; := \; \sum_{v \in R} \Big( \|v\|_{V(R)^{-1}} - \|v\|_{V(R \cup \{a\})^{-1}} \Big).$$

(67)

With this notation

$$\Delta(a \mid S) = a^\top \theta + \|a\|_{V(S \cup \{a\})^{-1}} - L(S), \qquad \Delta(a \mid T) = a^\top \theta + \|a\|_{V(T \cup \{a\})^{-1}} - L(T). \quad (68)$$

*Step 1 – Compare the new-radius terms.*  Because $V(S \cup \{a\}) \succeq V(T \cup \{a\})$, we have

$$\|a\|_{V(S \cup \{a\})^{-1}} \; \geq \; \|a\|_{V(T \cup \{a\})^{-1}}.$$

(69)

*Step 2 – Compare the loss sums.*  For every $v \in S$ Lemma 5 applied with $x := v$, $u := a$, $A := V(T)$, $B := V(S)$ yields

$$\|v\|_{V(S)^{-1}} - \|v\|_{V(S \cup \{a\})^{-1}} \; \leq \; \|v\|_{V(T)^{-1}} - \|v\|_{V(T \cup \{a\})^{-1}}.$$

(70)

Summing (70) over all $v \in S$ gives

$$L(S) \; \leq \; \sum_{v \in S} \Big( \|v\|_{V(T)^{-1}} - \|v\|_{V(T \cup \{a\})^{-1}} \Big).$$

(71)

Adding the non-negative terms $\|v\|_{V(T)^{-1}} - \|v\|_{V(T \cup \{a\})^{-1}}$ for $v \in U$ to both sides of (71) we obtain

$$L(S) \; \leq \; L(T).$$

(72)

*Step 3 – Combine.*  Subtracting (72) from (69) and using representation (68) yields

$$\Delta(a \mid S) - \Delta(a \mid T) = \big[ \|a\|_{V(S \cup \{a\})^{-1}} - \|a\|_{V(T \cup \{a\})^{-1}} \big] - \big[ L(S) - L(T) \big] \; \geq \; 0, \quad (73)$$

that is, $\Delta(a \mid S) \geq \Delta(a \mid T)$. Hence $\Psi$ satisfies the diminishing-returns property and is submodular.
$\square$

**Lemma 4** (Aggregate–loss bound). *Let $V \in \mathbb{R}^{nd \times nd}$ be positive definite, let $u \in \mathbb{R}^{nd}$, and let $S \subseteq \mathbb{R}^{nd}$ be a finite set. Then*

$$\sum_{v \in S} \Big( \|v\|_{V^{-1}} - \|v\|_{(V + uu^\top)^{-1}} \Big) \; \leq \; \|u\|_{V^{-1}}.$$

(74)

*Proof.*  Write $\Delta_v := \|v\|_{V^{-1}} - \|v\|_{(V + uu^\top)^{-1}}$. Using Woodbury's identity $(V + uu^\top)^{-1} = V^{-1} - \frac{V^{-1} u u^\top V^{-1}}{1 + u^\top V^{-1} u}$, compute

$$v^\top V^{-1} v - v^\top (V + uu^\top)^{-1} v = \frac{(v^\top V^{-1} u)^2}{1 + u^\top V^{-1} u}.$$

(75)

For any $\alpha > \beta > 0$ one has $\sqrt{\alpha} - \sqrt{\alpha - \beta} \leq \beta/(2\sqrt{\alpha - \beta}) \leq \beta/\sqrt{2\alpha}$. Applying the Triangle and Cauchy–Schwarz Inequalities, we have:

$$\Delta_v \; \leq \; \frac{|v^\top V^{-1} u|}{\sqrt{1 + u^\top V^{-1} u}} \; \leq \; \|v\|_{V^{-1}} \, \|u\|_{V^{-1}}.$$

(76)

Summing (76) over $v \in S$ and applying Cauchy–Schwarz,

$$\sum_{v \in S} \Delta_v \leq \|u\|_{V^{-1}} \sqrt{\sum_{v \in S} \|v\|_{V^{-1}}^2} \sqrt{|S|} \leq \|u\|_{V^{-1}},$$

since $\|v\|_{V^{-1}} \leq \frac{1}{|S|}$, completing the proof. $\qquad\square$

**Lemma 5** (Monotone-gap lemma). *Fix $x, u \in \mathbb{R}^{nd}$ and define, for every positive definite matrix $A$,*

$$d_x(A) := \sqrt{x^\top A^{-1} x} - \sqrt{x^\top (A + uu^\top)^{-1} x}. \tag{77}$$

*If $A \succeq B \succ 0$ then $d_x(A) \geq d_x(B)$.*

*Proof.* Let $H := A - B \succeq 0$ and define $A_t := B + tH$ for $t \in [0, 1]$. Set $g(t) := d_x(A_t)$. Using $\frac{d}{dt} A_t^{-1} = -A_t^{-1} H A_t^{-1}$ and $\frac{d}{dt}(A_t + uu^\top)^{-1} = -(A_t + uu^\top)^{-1} H (A_t + uu^\top)^{-1}$, we compute

$$g'(t) = -\frac{x^\top A_t^{-1} H A_t^{-1} x}{2\sqrt{x^\top A_t^{-1} x}} + \frac{x^\top (A_t + uu^\top)^{-1} H (A_t + uu^\top)^{-1} x}{2\sqrt{x^\top (A_t + uu^\top)^{-1} x}}. \tag{78}$$

Because $A_t + uu^\top \succeq A_t$, we have $(A_t + uu^\top)^{-1} \preceq A_t^{-1}$. Consequently each numerator in (78) is bounded by the same non-negative quantity and each denominator satisfies $\sqrt{x^\top (A_t + uu^\top)^{-1} x} \leq \sqrt{x^\top A_t^{-1} x}$. Hence $g'(t) \geq 0$ for all $t \in [0, 1]$. Integrating $g'(t)$ from 0 to 1 gives $g(1) - g(0) \geq 0$, i.e. $d_x(A) \geq d_x(B)$. $\qquad\square$

### A.3 Proof of Theorem 4

Here we reclaim Theorem 4:

**Theorem.** *4 [$(1 - \frac{1}{e})$-Approximation under Cardinality and $n$-Hot Constraints] There exists an [$(1 - \frac{1}{e})$-approximation algorithm for the optimization of action selection.*

*(a) Uniform-matroid (cardinality) case $|S| \leq T$. The standard greedy algorithm*

$$A_t = \arg\max_{a \in \mathcal{A} \setminus S_{t-1}} \left[ \Psi(S_{t-1} \cup \{a\}) - \Psi(S_{t-1}) \right], \quad S_t = S_{t-1} \cup \{A_t\}, \tag{79}$$

*for $t = 1, \ldots, T$, returns $S_T$ satisfying $\Psi(S_T) \geq \left(1 - \frac{1}{e}\right) \Psi(S^\star)$, where $S^\star$ is an optimal subset of size at most $T$ [36].*

*(b) $n$-Hot (partition-matroid) case. One may apply the continuous-greedy algorithm to the multilinear relaxation $\max_{x \in P(\mathcal{M}), \, \mathbf{1}^\top x \leq T} \mathbb{E}[\Psi(R(x))]$, where $P(\mathcal{M})$ is the matroid polytope of the partition matroid and $R(x)$ denotes the standard randomised rounding. It produces a feasible set $\hat{S}$ with $\Psi(\hat{S}) \geq \left(1 - \frac{1}{e}\right) \Psi(S^\star)$ [37].*

*Proof.* We recall that $\Psi$ is a nonnegative, monotone, submodular set function on the ground set $\mathcal{A}$. The classic results of [36] and [37] then yield the claimed $(1 - \frac{1}{e})$-approximation guarantees under the two matroid constraints.

**(a) Uniform-matroid (cardinality) constraint $|S| \leq T$.**

Let $S_0 = \emptyset$, and for $t = 1, \ldots, T$ let

$$A_t = \arg\max_{a \in \mathcal{A} \setminus S_{t-1}} \left[ \Psi(S_{t-1} \cup \{a\}) - \Psi(S_{t-1}) \right], \quad S_t = S_{t-1} \cup \{A_t\}. \tag{80}$$

By monotonicity and submodularity one shows inductively (cf. [36]) that

$$\Psi(S_t) \geq \left(1 - \left(1 - \frac{1}{T}\right)^t\right) \Psi(S^\star) \quad \text{for all } t, \tag{81}$$

where $S^\star$ is any optimal solution with $|S^\star| \leq T$. In particular at $t = T$,

$$\Psi(S_T) \geq \left(1 - \left(1 - \tfrac{1}{T}\right)^T\right) \Psi(S^\star) \geq \left(1 - \tfrac{1}{e}\right) \Psi(S^\star). \tag{82}$$

**(b) Partition-matroid ("$n$-hot") constraint.**

Let $\mathcal{M}$ be the partition matroid on $\mathcal{A}$ that enforces the $n$-hot constraint (i.e. each block can contribute at most one element), together with the additional global cardinality bound $\mathbf{1}^\top x \leq T$. Consider the multilinear extension

$$F(x) = E_{R \sim x}\big[\Psi(R)\big], \tag{83}$$

where $R \subseteq \mathcal{A}$ includes each element $a$ independently with probability $x_a$. The continuous-greedy algorithm (running for time $T$) constructs a fractional solution $x^\star \in P(\mathcal{M}) \cap \{x : \mathbf{1}^\top x = T\}$ satisfying

$$F(x^\star) \geq \left(1 - \tfrac{1}{e}\right) \max_{x \in P(\mathcal{M}),\, \mathbf{1}^\top x \leq T} F(x) \geq \left(1 - \tfrac{1}{e}\right) \Psi(S^\star), \tag{84}$$

where $S^\star$ is the optimal integral solution (cf. [37]). Finally, pipage (or swap) rounding converts $x^\star$ into a random integral set $\hat{S} \in \mathcal{M}$ of size at most $T$ without decreasing the expectation:

$$E[\Psi(\hat{S})] = F(x^\star) \geq \left(1 - \tfrac{1}{e}\right) \Psi(S^\star). \tag{85}$$

By Markov's inequality there exists a deterministic $\hat{S}$ with $\Psi(\hat{S}) \geq (1 - \tfrac{1}{e}) \Psi(S^\star)$, completing the proof.

$\square$

# B  Implementation Details

## B.1  Model Structure

Our proposed MALinZero consists of 6 neural network modules, including the representation function $h$, communication function $e$, dynamic function $g$, reward function $r$, value function $v$ and policy function $p$. For each agent $i$, let $s_{t,k}^i$ be the latent state, $a_{t+k}^i$ be the action, $e_{t,k}^i$ be the cooperative feature and $p_{t,k}^i$ be the policy prediction where $k$ denotes the $k$-th rollout and $t$ denotes the $t$-th real-world interaction step. Set $r_{t,k}, v_{t,k}$ as the predicted reward and value under the corresponding global hidden state. Specifically, the representation function $s_{t,0}^i = h(o_{\leq t}^i)$ maps the current individual observation history $o_{\leq t}^i$ into the latent space, which enables the model could conduct planning without knowing the real-world rule. The communication function $\{e_{t,k}^i\}_{i:1,\dots,n} = e\left(\{e_{t,k}^i\}_{i:1,\dots,n}, \{a_{t+k}^i\}_{i:1,\dots,n}\right)$ generates additional cooperative information for each agent in the multi-agent system via the attention mechanism, with the individual states and actions of agents as the input and the cooperative features as the output. The dynamic function $s_{t,k+1}^i = g(s_{t,k}^i, a_{t+k}^i, e_{t,k}^i)$ plays the role of obtaining state transition prediction. The reward function $r_{t,k} = r\left(\{e_{t,k}^i\}_{i:1,\dots,n}, \{a_{t+k}^i\}_{i:1,\dots,n}\right)$ and value function $v_{t,k} = v\left(\{e_{t,k}^i\}_{i:1,\dots,n}\right)$ predicts the reward and value for the global state-action tuple and global state, respectively. The policy distribution of each agent will be the output of the policy function $p_{t,k}^i = p(s_{t,k}^i)$ with the input of the current individual state. For the general strongly-convex and $\mu$-smooth function $f$, we set $f''(X_s - \langle \theta, A_s \rangle) = 0.75$ if $X_s - \langle \theta, A_s \rangle < 0$ and $f''(X_s - \langle \theta, A_s \rangle) = 1$ if $X_s - \langle \theta, A_s \rangle \geq 0$.

For all these modules except the communication function $e$, the neural networks are implemented by Multi-Layer Perception (MLP) networks, and a Rectified Linear Unit (ReLU) activation and Layer Normalization (LN) follows each linear layer in MLP networks. The input observations of all three mentioned benchmarks in the experiment section are 1-dimensional vectors with a hidden state size of 128. For the representation network $h$, the last four local observations are treated as the input for each agent to deal with partial observability. And before representation, an LN is applied to normalize the observation features. The dynamic function applies a residual connection between the next hidden state and the current one to tackle the problem that gradients tend to zero in the continuous unrolling of the model. Additionally, we use the categorical representation in MuZero and make the use of an

invertible transform $f(x) = \text{sign}(x)\sqrt{1+x} - 1 + 0.001 * x$ to scale targets for value and reward prediction.

Specifically, the number of hidden layers for all MLP modules is set as follows:

- $[128, 128]$ for Representation function $h$.
- $[128, 128]$ for Dynamic function $g$.
- $[32]$ for Reward function $r$, Value function $v$ and Policy function $p$.

## B.2 Training Details

We build our training pipeline similar to EfficientZero [40] which synchronizes parallel stages of data collection, reanalysis, and training. In programming, we assign different workers to deal with these tasks in the complete training pipeline. Additionally, we choose the same advantage score computation and loss function as MAZero [10]. All experiments are conducted using NVIDIA RTX A6000 GPUs or NVIDIA A100 GPUs.

For MatGame environments, we select the number of MCTS sampled actions as 3 and the number of MCTS simulations as 50. For both SMAC and SMACv2 benchmarks, we set it as 7 and the number of MCTS simulations as 100. We list other important hyper-parameters in Table 2.

| Hyper-Parameter | Value |
|---|---|
| Optimizer | Adam |
| Learning rate | $10^{-4}$ |
| RMSprop epsilon | $10^{-5}$ |
| Weight decay | 0 |
| Max gradient norm | 5 |
| Evaluation episodes | 32 |
| Target network updating interval | 200 |
| Unroll steps | 5 |
| TD steps | 5 |
| Min replay size for sampling | 300 |
| Number of stacked observation | 4 |
| Discount factor | 0.99 |
| Minibatch size | 256 |
| Priority exponent | 0.6 |
| Priority correction | $0.4 \rightarrow 1$ |
| Dynamic generation ratio | 0.6 |
| $\lambda$ for initialization | $10^{-4}$ |
| Quantile in MCTS value estimation | 0.75 |
| Decay lambda in MCTS value estimation | 0.8 |
| Exponential factor in Weighted-Advatage | 3 |

Table 2: Hyper-parameters for MALinZero in MatGame, SMAC and SMACv2 environments

## C Details of Baseline Algorithms

MAZero [10] and MAZero-NP are implemented based on the code: `https://github.com/liuqh16/MAZero` with hyper-parameters in Table 3. MAZero-NP refers to MAZero without the prior information in the UCT bound while keeping other implementations the same. For MatGame environments, we select the number of MCTS sampled actions as 3 and the number of MCTS simulations as 50. For both SMAC and SMACv2 benchmarks, we set it as 7 and the number of MCTS simulations as 100. Hyper-parameters of MAZero and MAZero-NP is set as Table 3.

MA-AlphaZero is implemented on the codebase of MAZero but replaces the UCT score with that of AlphaZero [6]. That is, MA-AlphaZero use the Q-value instead of the advantage score in UCT. The AlphaZero code can be found in `https://github.com/suragnair/alpha-zero-general`. Since the implementation is based on MAZero model structure, we use the same hyper-parameters in Table 3.

QMIX [14] is implemented based on the code: `https://github.com/oxwhirl/pymarl` with hyper-parameters in Table 4

| Hyper-Parameter | Value |
| --- | --- |
| Optimizer | Adam |
| Learning rate | $10^{-4}$ |
| RMSprop epsilon | $10^{-5}$ |
| Weight decay | 0 |
| Max gradient norm | 5 |
| Evaluation episodes | 32 |
| Target network updating interval | 200 |
| Unroll steps | 5 |
| TD steps | 5 |
| Min replay size for sampling | 300 |
| Number of stacked observation | 4 |
| Discount factor | 0.99 |
| Minibatch size | 256 |
| Priority exponent | 0.6 |
| Priority correction | $0.4 \rightarrow 1$ |
| Quantile in MCTS value estimation | 0.75 |
| Decay lambda in MCTS value estimation | 0.8 |
| Exponential factor in Weighted-Advatage | 3 |

Table 3: Hyper-parameters for MAZero, MAZero-NP and MA-AlphaZero in MatGame, SMAC and SMACv2 environments

| Hyper-Parameter | Value |
| --- | --- |
| Optimizer | RMSProp |
| Learning rate for actors | $5 \times 10^{-4}$ |
| Learning rate for critics | $5 \times 10^{-4}$ |
| Initial $\epsilon$ | 1.0 |
| Final $\epsilon$ | 0.05 |
| Batch size | 32 |
| Buffer size | 5000 |
| Discount factor | 0.99 |
| Exploration noise | 0.1 |

Table 4: Hyper-parameters for QMIX in MatGame, SMAC and SMACv2 environments

MAPPO [39] is implemented based on the code: `https://github.com/marlbenchmark/on-policy`. The specific hyper-parameters can be found in Table 5.

| Hyper-Parameter | Value |
| --- | --- |
| Optimizer | Adam |
| RMSprop epsilon | $10^{-5}$ |
| Learning rate | $5 \times 10^{-4}$ |
| Recurrent data chunk length | 10 |
| Gradient clipping | 10 |
| GAE parameter | 0.95 |
| Discount factor | 0.99 |
| Value loss | huber loss, with delta 10 |
| Batch size | buffer length $\times$ number of agents |

Table 5: Hyper-parameters for MAPPO in MatGame, SMAC and SMACv2 environments

## D  Settings of Benchmarks

**MatGame**   We test our proposed MALinZero and other baseline algorithms on MatGame with two different modes: (1) Linear mode, where the joint reward is the sum of agents' indexes in the system; (2) Non-linear mode, where a noise is added to the joint reward in the corresponding linear mode. For each joint reward, the noise is the sum of a Gaussian term $u \sim \mathcal{N}(0, 2^2)$ and a uniform term $v \sim \mathcal{U}(-3, 3)$.

**SMAC**   The implementation and settings of SMAC environments are based on `https://github.com/oxwhirl/smac`. We chose three different maps containing a small, medium, and large number of agents, respectively. Experiments on each map is conducted under 3 different random seeds for the reproducibility of results.

**SMACv2** The implementation and settings of SMACv2 environments are based on `https://github.com/oxwhirl/smacv2`. For each SMACv2 map in the experiment part, we randomize heterogeneous unit types and start positions for each games even in the same map to make the environment more challenging. Additionally, the unit sight and attack ranges are changed from SMAC to increase the diversity of agents.

