# OpenReview forum: "MALinZero: Efficient Low-Dimensional Search for Mastering Complex Multi-Agent Planning"
_NeurIPS.cc/2025/Conference — NeurIPS 2025 poster_

### Official Review · Reviewer_EsMc · 2025-06-22

**Clarity:** 2
**Significance:** 3
**Originality:** 3
**Rating:** 4
**Confidence:** 3

**Summary:**

This paper, MALinZero, proposes a new way to tackle a big problem in multi-agent planning with Monte Carlo Tree Search (MCTS): the action space gets huge very fast. Their idea is to represent joint-action returns in a low-dimensional space, treating it as a contextual linear bandit problem. They introduce a new LinUCT (Linear Upper Confidence Bound applied to trees) for MCTS that helps explore and exploit this lower-dimensional space. They also use a special "loss function" that's convex and smooth to make sure they don't miss out on good actions. The paper claims MALinZero has good theoretical guarantees, like regret bounds, and they show that picking actions is like maximizing a submodular function, which has a known approximation algorithm. Plus, they say they made the back-propagation part more efficient. They tested MALinZero on matrix games, SMAC, and SMACv2. They report it beats other model-based and model-free methods, learning faster and performing better.

**Questions:**

1. **How are "latent per-agent action rewards" learned?** Section 3.1 talks about these rewards, $\theta^{\ast}$, but the high-level representation model $h_{\theta}$ (mentioned for MuZero) isn't clearly connected to how these specific latent rewards are actually derived from observations. Is there a neural network learning this projection? If so, what's its structure?
2. **Details on the General Loss Function $f$:** You introduce the generalized loss function $f$ with $w_{+}$ and $w_{−}$. How are these values determined for different rewards ($X_s\ge\langle\theta,A_s\rangle$ vs. $X_s<\langle\theta,A_s\rangle$)? Are they fixed, hand-tuned hyperparameters, or are they learned during training? This is important for practical application.
3. **Clarify Back-Propagation Notation in Algorithm 1:** Please clarify the $\theta(s)$ update: "$\theta(s)\leftarrow V(s)^{-1}r_{k}a$". How does this specific $r_{k}$ relate to $X_s$ in the sum for $\hat{\theta}_{t}$? A consistent and precise update rule would be much better.
4. **More in Main Text, Less in Appendices, please!** The paper refers to Appendices A, B, and D constantly. Without them, it's impossible to properly evaluate the theoretical claims, understand the model's architecture, or verify the experimental setup. Could you explain them more clearly in the MAIN TEXT, or at least provide a sketch explanation?

My score could increase if you address my concerns and clarify the core mechanisms (representation learning, loss function parameters, back-propagation notation). It would decrease if these fundamental aspects remain unclear.

**Ethical Concerns:**

["NO or VERY MINOR ethics concerns only"]

**Final Justification:**

The authors resolved most of my concerns and questions about the technical details of MALinZero in the rebuttal, and in light of this, I have raised my score towards acceptance. However, the clarity of its writing still needs improvement, and I can only believe that the authors will fulfill the promised revisions.

**Limitations:**

The authors mention two limitations: the current linear bandit formulation and the open problem of fully decomposable representations. These are fair.

However, I think they should also explicitly acknowledge:

- **Computational Overhead of Submodular Maximization:** While theoretically elegant, the practical speed implications of solving a submodular maximization problem (even approximately) during each MCTS selection step need to be considered and perhaps quantified, especially for time-sensitive applications.
- **Hyperparameter Sensitivity of the Generalized Loss Function:** The novel loss function $f$ with its potentially tunable parameters ($w_{+}$, $w_{−}$) could introduce new challenges for hyperparameter tuning. It would be good to discuss the sensitivity of the method to these parameters as a potential limitation.

**Paper Formatting Concerns:**

There are no major formatting issues in this paper.

**Quality:**

2

**Strengths And Weaknesses:**

### Strengths

- **Tackles a Real Problem:** They're going after the exponential action space issue in multi-agent MCTS, which is a genuinely tough nut to crack. That's a solid motivation for the work.
- **Novel Approach for MCTS:** Using a contextual linear bandit and LinUCT to project into a low-dimensional space for *multi-agent MCTS* is pretty neat. While these ideas exist elsewhere (like low-dimensional representations in MARL), their specific combination and theoretical backing for MCTS exploration is a notable contribution.
- **Theoretical Backing:** They've got regret bounds and they've shown the action selection is a submodular optimization problem, which gives it a good theoretical foundation.
- **Good Experimental Results:** The empirical results look strong across the benchmarks they chose. Outperforming existing methods, especially on SMACv2 where they claim to nearly double winning rates, is impressive. The speed-up claims are also interesting.

### Weaknesses

- **The "General $f$" Loss Function Needs More Detail:** They introduce this fancy convex and smooth loss function $f$ to avoid underestimating good actions. They give an example with $w_{+}$ and $w_{−}$. But how do you *choose* these weights in practice? Is it fixed? Is it tuned? Is it learned? This is crucial for understanding its real-world applicability and why it helps so much in the ablation study.
- **Dynamic Node Generation (DNG) Sampling:** Algorithm 1 says DNG uses "$\beta$ and $P$ as Sampled MuZero" for sampling new nodes. That's pretty vague. What exactly are $\beta$ and $P$ in their context? How does this sampling work? Clarity here is needed.
- **Computational Overhead of Submodular Maximization:** They proved action selection is submodular, which is cool, but they don't really talk about the practical speed impact of using an approximation algorithm for it during the MCTS rollout. MCTS relies on fast tree traversal. Does this add a noticeable overhead? How does it compare to the simpler UCB-like selections in other MCTS methods?
- **Hard to Follow WITHOUT Appendix:** It's impossible to properly review this paper without the appendices. They refer to them *constantly* for proofs, architecture details, and experimental settings. Without these, I can't check their math, understand their "low-dimensional representation" fully, or see how their models are structured.

---

> ### Author Rebuttal · Authors · 2025-07-30
>
> Thank you for your constructive and detailed suggestions. Our answers to your questions are provided below.
>
> Due to character limit, in order to use more space to answer your concerns in detail, we will not repeat the content of your question here, but will use numbers instead. We are deeply sorry for the inconvenience caused.
>
> **W1**
>
> **Q2**
>
> In the current version of MALinZero, the values of \\(w_+\\) and \\(w_-\\) are fixed as \\(1\\) and \\(0.75\\) respectively. The design and setting of weights are inspired by the classic multi-agent reinforcement learning algorithm WQMIX [1]. In WQMIX, two fixed different weights are applied for cases where the joint reward is over-estimated and under-esitimated. We are interested in investigating whether a more sophisticated weighting function will bring extra performance improvement empirically and plan to do this in the future work. From the theoretical aspect, we have analyzed and derived the regret of LinUCT with the strongly-convex and \\(\mu\\)-smooth function \\(f\\) so the difference of function designing will not damage our theoretical results.
>
> **W2**
>
> \\(\beta\\) and \\(P\\) are applied in the Expansion stage. \\(P(s,a)\\) is the policy distribution output of policy network for state \\(s\\) and action \\(a\\). Both of them provide the prior information to guide the sampling in Expansion to pay more attention to actions with higher policy priors. Specifically, we compute a vector \\(\beta_i(s)\\) for agent \\(i\\) and \\(\beta_i(s)\propto \operatorname{pow}(P(s), 1/\tau)\\) where \\(\tau\\) is the temperature coefficient. In implementation, we sample the individual action with \\(\beta_i\\) values for each action and then concatenate them to obtain the joint actions. By repeating this process for \\(K\\) times, we can get the \\(K\\) child nodes for the current node.
>
> **W3**
>
> This is a great question. Putting our short answers first: Yes, Submodular Maximization will introduce minor computation overhead, but it is necessary to leverage the linear factorization in MALinZero to reduce the search complexity of joint actions from the standard \\(\mathcal{O}(d^n)\\) to \\(\mathcal{O}(dn)\\). The overhead of Submodular Maximization is negligible with respect to this reduction from exponential to linear.
>
> With the linear structure in LinUCB, the problem of computing confidence bound and selecting optimal actions becomes a linear contextual bandit problem that cannot be computed using standard UCB. Thus the Submodular Maximization is proposed to solve this new problem. Running the LinUCB algorithm will incur minor additional cost. However, the computation leverages a linear structure with sampling \\(\mathcal{O}(dn)\\) actions rather than the standard \\(\mathcal{O}(d^n)\\). Compared with this reduction from exponential to linear, the additional computation cost is negligible. Our evaluation shows that the computational cost is comparable to that of the MAZero [4] method. We will include additional explanations in the revised version.
>
> We also provide further details on this computation. First, we only select actions requiring submodular maximization for \\((1-\zeta)\chi\\) where \\(\zeta\\) is the dynamic generation ratio and \\(\chi\\) is the maximum number of child nodes for a node. For example, if we expand \\(5\\) child nodes at most for a node and set the dynamic generation ratio \\(\zeta\\) as \\(0.6\\), then we only need to select \\(2\\) action by solving the submodualr maximization thus the overhead is not that huge but the benefit is noticeable. If we search the currently optimal action by the exhaustive method, the complexity will be \\(\mathcal{O}(d^n)\\) where \\(d\\) is the size of individual action space and \\(n\\) is the agent number. However, the complexity of submodular maximization is \\(\mathcal{O}(n^2d)\\), which is much smaller than that of exhaustive method.
>
> **W4**
>
> **Q4**
>
> Unfortunately we were not able to include all details in the main text due to page limit. We will make some room and provide sketch explanations in the main text, about the theoretical results, model's architecture and the experiment setting in the main text. Please see the following text.
>
> __Model architecture.__
> MALinZero consists of 6 neural networks to be learned during the training and the parameter \\(\theta\\) is to be estimated from initialization for a single MCTS process. Specifically, with network parameter \\(\phi\\), there are 6 key functions: the representation function \\(s^i_{t,0}=h_\phi(o^i_{\le t})\\) that maps the current individual observation history into the latent space, the communication function $e^1_{t,k},\dots, e^n_{t,k} = e_\phi(s^1_{t,k},\dots,s^n_{t,k},a^1_{t,k},\dots,a^n_{t,k})$ that generates cooperative information for each agent via the attention mechanism, the dynamic function \\(s^i_{t,k+1} = g_\phi(s^i_{t,k}, a^i_{t+k}, e^i_{t,k})\\) that plays the role of transition function, the reward function \\(r_{t,k}=r_\phi (s^1_{t,k},\dots,s^n_{t,k},a^1_{t,k},\dots,a^n_{t,k})\\) and the value function \\(v_{t,k}=v_\phi(s^1_{t,k},\dots,s^n_{t,k})\\) that predicts the reward and value respectively, and the policy function \\(p^i_{t,k} = p_\phi(s^i_{t,k})\\) that predicts the policy distribution for the given state. The subscript \\(k\\) denotes the index of unrolling steps within one simulation from the root node in MCTS. The update of estimated \\(\theta\\) takes place in the Back-propagation stage and the detailed process is analyzed in the main text, especially in the last part of Section 3.3. For all these modules except for the communication function \\(e_\phi\\), the neural networks are implemented by Multi-Layer Perception (MLP) networks and a Rectified Linear Unit (ReLU) activation and Layer Normalization (LN) follows each linear layer in MLP networks. The Transformer architecture [2] with three stacked layers to encode the state-action pairs. And agents process local dynamics and make predictions with the encoded information.
>
> __Experiment setting.__
> All experiments are conducted using NVIDIA RTX A6000 GPUs and NVIDIA A100 GPUs. For MatGame environments, the number of sampled actions for each node in MCTS is 3 and the number of MCTS simulations is 50. For both SMAC and SMACv2 benchmarks, we set them as 7 and 100, respectively. We build our training pipeline similar to EfficientZero [3] which synchronizes parallel stages of data collection, reanalysis, and training.
>
> __Proof outline.__ We have provided a brief outline in the appendix at the beginning of each proof. We will further simplify and move some of these to the main text. Thanks for this suggestion!
>
> **Q1**
>
> Our solution is not try to learn the "per-agent action rewards". It learns the joint reward, and more importantly, the corresponding confidence bound under a linear structure. For each node \\((s,a)\\) where \\(s\\) denotes the state and \\(a\\) denotes the action leading to \\(s\\), $\theta^\*\in\mathbb{R}^{nd}$ is maintained and learned during the MCTS process. The linear structure allows us to learn the confidence bounds (by estimating \\(\theta^\*\\) and its confidence set \\(\mathcal{C}_t\\) on Page 4) and directly use the results for action selection (by solving the Submodular Maximization on Page 5). Our solutions combines these steps into a single process to sample actions using the confidence bounds.
>
> To provide further details, when a new node is expanded and added to the search tree, $\hat\theta$ used to estimate \\(\theta^\*\\) is initialized as \\(\mathbf{0}\\). The per-node initialization makes sense because it is helpful to achieve better per-state reward estimation and the next-node selection for each node can be viewed as an independent bandit problem. \\(\hat\theta\\) is update in the Back-propagation stage by the observed reward. After multiple round of update, the estimator will be more and more precise.
>
> The representation model \\(h_\theta\\) is only applied in the root node to project the real observation history \\(o^i_{\le t}\\) until time step \\(t\\) of agent \\(i\\) into an individual latent state \\(s^i_t = h_\theta(o^i_{\le t})\\). Thus, all models (dynamic model, policy model, etc.) in MCTS are learned and executed in the latent space. The input of \\(h_\theta\\) is the local observation history and the output is the state in the latent space. \\(h_\theta\\) is implemented as Multi-Layer Perception (MLP) networks, where each linear layer in MLP is followed by a Rectified Linear Unit (ReLU) activation and a Layer Normalisation (LN) layer. Specifically, we use the hidden state size of 128 and the hidden layers for the MLP applied in \\(h_\theta\\) is [128, 128].
>
> **Q3**
>
> \\(r_k\\) in the Algorithm 1 refers to the observed reward \\(X_s\\) after taking action \\(a\\) at the \\(k\\)-th node in the search path. We use the notation \\(r_k\\) here since \\(s\\) in Algorithm 1 \\(s\\) refers to the state while in Section 3.1 \\(s\\) of \\(X_s\\) means a time step. We will check the usage precision of notations through the paper to improve the readability.
>
>
> [1] Rashid, Tabish, et al. "Weighted qmix: Expanding monotonic value function factorisation for deep multi-agent reinforcement learning." Advances in neural information processing systems 33 (2020): 10199-10210.
>
> [2] Vaswani, Ashish, et al. "Attention is all you need." Advances in neural information processing systems 30 (2017).
>
> [3] Ye, Weirui, et al. "Mastering atari games with limited data." Advances in neural information processing systems 34 (2021): 25476-25488.
>
> [4] Liu, Qihan, et al. "Efficient multi-agent reinforcement learning by planning." arXiv preprint arXiv:2405.11778 (2024).

---

> > ### Comment · Reviewer_EsMc · 2025-08-06
> >
> > Thank you for the detailed rebuttal and for clarifying several key points. The explanations address most of my initial concerns, particularly regarding the implementation details and theoretical claims.
> >
> > The commitment to including the appendices, with a sketch of the proofs, model architecture, and experimental settings in the main text, is encouraging. This will significantly improve the paper's clarity, reproducibility, and overall quality.
> >
> > Given these comprehensive responses, I believe the paper is much stronger. If the authors can fulfill all the revisions promised in this rebuttal, this work will be a valuable contribution to the multi-agent MCTS community. I will raise my score accordingly.

---

> > > ### Author Response · Authors · 2025-08-07
> > >
> > > Thanks for your detailed review and comments! We will fulfill the revisions in the rebuttal and include more contents in the appendices into the main text.

---

> ### Comment · Area_Chair_kuEJ · 2025-08-05
>
> Thanks to the authors for their rebuttal! Since this paper got borderline reviews, a critical evaluation whether the rebuttal addresses the reviewer's questions and criticisms is very important. Could the reviewer please check whether the rebuttal still left them with open questions, whether their concerns are alleviated enough to raise their score, or just acknowledge that neither is the case?

---

### Official Review · Reviewer_PKqZ · 2025-06-30

**Clarity:** 2
**Significance:** 2
**Originality:** 3
**Rating:** 4
**Confidence:** 3

**Summary:**

This paper proposes MALinZero for efficient low-dimensional search in multi-agent planning. The approach projects joint-action returns into a low-dimensional space via a contextual linear bandit formulation, and solves this contextual linear bandit problem with convex and $\mu$-smooth loss functions. It reduces the tree search complexity from the exponential in the number of agents to linear in the number of agents. Results on SMAC demonstrate that MALinZero outperforms state-of-the-art methods in this multi-agent tasks.

**Questions:**

- Are all agents considered within a single MCTS process? If so, is MCTS applied during test time? If it is used, does the method still comply with the Centralized Training and Decentralized Execution (CTDE) paradigm?
- The paper introduces a strongly convex and $\mu$-smooth distance measure f, and discusses its benefits. However, it is unclear how f achieves the desired behavior. For example, line 186 states: “For higher observed rewards X_t that are likely optimal, the distance measure f will have a larger acceleration”. Does this behavior also apply to higher observed rewards that are not optimal? Further clarification and justification would strengthen the argument.
- In high-dimensional multi-agent tasks, will the same state be encountered multiple times? I’m curious whether the MCTS is truly maintaining a tree structure, or just some independent trajectories, especially in this multi-agent setting.
- see Weaknesses.

**Ethical Concerns:**

["NO or VERY MINOR ethics concerns only"]

**Final Justification:**

The authors have done a good job in addressing my concerns. I raise my score accordingly.

**Limitations:**

The authors acknowledge the limitations of their work and outline directions for future improvement.

**Quality:**

2

**Strengths And Weaknesses:**

Strengths
- The paper proposes a novel method for multi-agent planning that reduces the complexity of tree search from exponential to linear in the number of agents, which is theoretically sound.
- The proposed method is evaluated on the SMAC benchmark and compared against both model-free and model-based baselines.

Weaknesses
- The writing can be improved. There are many repeated phrases across the abstract, introduction, and related work sections, which makes the paper feel redundant.
- It would be helpful to clearly state at the beginning of the paper that the proposed method applies only to the cooperative multi-agent setting.
- The provided code repository lacks a README file, making it difficult to install dependencies and reproduce the results.

minors:
- line 149, linear bandit problem []
- line 178, UCB1 and LinUCB []

---

> ### Author Rebuttal · Authors · 2025-07-30
>
> Thank you for your detailed feedback. We have made corresponding revisions to the paper per the comments that reviewers kindly provided.
>
> **W1: The writing can be improved. There are many repeated phrases across the abstract, introduction, and related work sections, which makes the paper feel redundant.**
>
> We will improve some of the expressions in the later version of our paper and try our best to make it more concise and clearer for readers.
>
> **W2: It would be helpful to clearly state at the beginning of the paper that the proposed method applies only to the cooperative multi-agent setting.**
>
> Thanks for your suggestion. We will clarify that MALinZero applies to cooperative multi-agent settings.
>
> **W3: The provided code repository lacks a README file, making it difficult to install dependencies and reproduce the results.**
>
> Thanks for your understanding. Due to the latest rebuttal format, we can not include any links or PDF files during this process, so it is hard to provide a clear README file now. The README file will be updated to our supplementary file later.
>
> **Q1: Are all agents considered within a single MCTS process? If so, is MCTS applied during test time? If it is used, does the method still comply with the Centralized Training and Decentralized Execution (CTDE) paradigm?**
>
> We consider a single MCTS process.
>
> During the test time, each agent will run its local instance of MCTS with the learned networks to simulate a fixed number of rollouts and then select the action according to the policy distribution of the root node. During training, action selections after factorizing and computing local confidence bounds are decentralized. Since each agent runs its local MCTS, MALinZero is completely CTDE. Specifically, all models are learned in the centralized manner with global states while the execution is fully decentralized: each agent carries its own copy of the learned model and exchanges only the observation to others without the central controller.
>
>
> **Q2: The paper introduces a strongly convex and $\mu$-smooth distance measure $f$, and discusses its benefits. However, it is unclear how $f$ achieves the desired behavior. For example, line 186 states: “For higher observed rewards $X_t$ that are likely optimal, the distance measure f will have a larger acceleration”. Does this behavior also apply to higher observed rewards that are not optimal? Further clarification and justification would strengthen the argument.**
>
> Intuitively, the introduction of a distance measure $f$ exploits the fact that estimation errors on different actions do not have the same impact on performance. Overestimating the return of an optimal action would still allow it to be selected (with no performance gap), while underestimating the return of a non-optimal action would not make it favorable (again with no performance gap). However, the opposite is not true (i.e., overestimating a non-optimal action or underestimating an optimal action).
>
> WQMIX [1] first leveraged this idea. Compared with QMIX [2], WQMIX does not treat each agent equivalently while it assigns higher weights for the joint actions with higher value to overcome the information loss of optimal actions due to the average operation in QMIX.
>
> Let's further analyze the example on line 191. Let \\(r=X_s-\langle\theta, A_s\rangle\\) and then we can get \\(\frac{\partial f(r)}{\partial \theta}=2 w \frac{\partial r}{\partial \theta}\\). When \\(r\ge 0\\), i.e., the model is under‑estimating, we assign\\(w=w_+\\) with a large \\(w_+\\) to amplify the update step \\(\Delta\theta\propto-2w_+r\partial r\\), thus accelerating the learning of \\(\theta\\). Otherwise, if \\(r < 0\\), we will assign a small weight \\(w_-\\) to \\(w\\) to avoid the over-correction since the model is over-estimating. And this behavior also applies to the higher observed rewards that are not optimal since \\(w\\) depends on whether we over- or under-estimate the reward.
>
>
> **Q3: In high-dimensional multi-agent tasks, will the same state be encountered multiple times? I’m curious whether the MCTS is truly maintaining a tree structure, or just some independent trajectories, especially in this multi-agent setting.**
>
> In MALinZero or many other search-based reinforcement learning algorithms [3] [4] [5], the node in a MCTS process is denoted by \\((s,a)\\) where \\(s\\) is the state and \\(a\\) is the action leading to this state. At a certain state \\(s\\), if the algorithm selects an action \\(a'\\), then the index will move from the node \\((s,a)\\) to node \\((s', a')\\) where \\(s'\\) comes from the dynamic model. During the entire MCTS process, every time \\(a'\\) is selected at \\((s,a)\\), the index will point to \\((s',a')\\). Meanwhile, the number of sampled actions is finite, so nodes are likely getting revisited within \\(K\\) simulation times -- which is the necessary to sample actions with not only a high observed return (due to limited samples), but also actions with high potential to yield a high return, through the computation of a confidence bound. Therefore, the MCTS is truly maintaining a tree structure.
>
> **Minors: line 149, linear bandit problem [], line 178, UCB1 and LinUCB []**
>
> We will revise the lost reference you mentioned in the later version of our paper, which should be [6].
>
> [1] Rashid, Tabish, et al. "Weighted qmix: Expanding monotonic value function factorisation for deep multi-agent reinforcement learning." Advances in neural information processing systems 33 (2020): 10199-10210.
>
> [2] Rashid, Tabish, et al. "Monotonic value function factorisation for deep multi-agent reinforcement learning." Journal of Machine Learning Research 21.178 (2020): 1-51.
>
> [3] Schrittwieser, Julian, et al. "Mastering atari, go, chess and shogi by planning with a learned model." Nature 588.7839 (2020): 604-609.
>
> [4] Hubert, Thomas, et al. "Learning and planning in complex action spaces." International Conference on Machine Learning. PMLR, 2021.
>
> [5] Liu, Qihan, et al. "Efficient multi-agent reinforcement learning by planning." arXiv preprint arXiv:2405.11778 (2024).
>
> [6] Li, Lihong, et al. "A contextual-bandit approach to personalized news article recommendation." Proceedings of the 19th international conference on World wide web. 2010.

---

> > ### Comment · Reviewer_PKqZ · 2025-08-05
> >
> > I appreciate the authors' efforts in addressing the concerns raised in the previous review. About Q3, do you have experiments to show the visit counts of each note, so that we can see how the visit counts are distributed? This will help to understand the effectiveness of your method.

---

> > > ### Author Response · Authors · 2025-08-06
> > >
> > > Thanks for your comment! In the algorithm implementation stage, we have tested the relationship between visit counts and node values and found that node with high values tend be visited more times, which demonstrated the effectiveness of MALinZero. Here we will show some experiment results on MatGame (6 agents, 8 actions, Linear and Non-Linear).
> > >
> > > | Agent | Action |    Type     | Node ID | Node Value | Visit Count |
> > > |:-----:|:------:|:-----------:|:-------:|:----------:|:-----------:|
> > > |   6   |   8    |   Linear    |   1-1   |   0.972    |     42      |
> > > |   6   |   8    |   Linear    |   1-2   |   0.961    |     30      |
> > > |   6   |   8    |   Linear    |   1-3   |   0.893    |     26      |
> > > |   6   |   8    |   Linear    |   1-4   |   0.841    |     15      |
> > > |   6   |   8    |   Linear    |   1-5   |   0.817    |      4      |
> > > |   6   |   8    | Non-Linear  |   2-1   |   0.961    |     44      |
> > > |   6   |   8    | Non-Linear  |   2-2   |   0.841    |     32      |
> > > |   6   |   8    | Non-Linear  |   2-3   |   0.824    |     27      |
> > > |   6   |   8    | Non-Linear  |   2-4   |   0.819    |     12      |
> > > |   6   |   8    | Non-Linear  |   2-5   |   0.798    |      6      |
> > >
> > > We set the simulation number as $50$ and the maximum child nodes as 3. We also normalize the node value for clear comparison in the table.  Due to the large amount of nodes in a tree, we only show the top 5 nodes with highest LinUCT values when the MCTS process is complete to improve the readability of this response.
> > >
> > > In MatGame (6,8,Linear), node 1-1 is the child node of the root and node 1-2 is the child node of 1-1. With the guidance of MCTS, node 1-3, 1-4 and 1-5 are respectively the child nodes of the former. This result demonstrates that LinUCT in MALinZero could help the search to find the optimal unrolling trajectory thus optimze the policy distribution. In MatGame (6, 8, Non-Linear), we also find that nodes with high values tend to be in the same subtree or one unrolling trajectory.

---

> > > > ### Comment · Reviewer_PKqZ · 2025-08-07
> > > >
> > > > I thank the authors for further clarifications. My concerns have been addressed. I rasised my score accordingly.

---

> > > > > ### Author Response · Authors · 2025-08-07
> > > > >
> > > > > Thanks for your feedback!

---

### Official Review · Reviewer_oPgB · 2025-07-02

**Clarity:** 2
**Significance:** 2
**Originality:** 3
**Rating:** 4
**Confidence:** 3

**Summary:**

The authors address the problem of the exponential action space blowup in multi-agent model based RL algorithms that rely on tree-search (like MAzero). The authors learn a low dimensional linear approximation of the reward function (for every hidden state).  The authors then propose using the UCB algorithm for a linear bandit, to select actions for expanding the tree search. This strikes a balance between exploration of new actions and exploitation of high rewarding actions, instead of expanding the tree for all possible actions. In benchmark tasks like MatGame and SMAC, the algorithm shows consistent improvement over baselines.

**Questions:**

1) Do you need to learn a reward function new approximation for different latent states? How computationally efficient is this? How is data generated to learn this function?

2) At every-step during expansion, you need to run the linUCB algorithm to select an action, what additional computational and memory overheads does this cause?

3) What happens when compute becomes orders of magnitude cheaper? Do you think the low rank approximation will still achieve better performance?

**Ethical Concerns:**

["NO or VERY MINOR ethics concerns only"]

**Final Justification:**

My final recommended score is 4. The algorithm solves the problem of large action spaces in multi-agent planning. They do so by learning a low dimensional linear approximation of the reward function and rapidly solving a contextual bandit problem to select actions for node expansion during tree search. The paper also demonstrates stronger performance in common multi agent RL benchmarks.

**Limitations:**

yes

**Paper Formatting Concerns:**

The paper formatting seems to be correct to me.

**Quality:**

3

**Strengths And Weaknesses:**

Strengths:
1) The strengths of the paper are a clear and detailed theoretical analysis of the MALinZero algorithm.
2) The empirical performance on all benchmark tasks considered seem strong when compared to the baselines.

Weaknesses:
1) The paper seems to have a lot of technical details about UCB, linear bandits and regret analysis. This makes it a little difficult to read for practitioners / non-theory researchers. It also makes it less clear if the main contributions are theoretical /  empirical. I do not work in multi-agent RL /  multi-agent RL theory, so it is a bit difficult to gauge how strong the theoretical / empirical results are.

2) How relevant do these algorithms remains when compute becomes orders of magnitude cheaper? Every agent can consider other agents as a part of the environment and use algorithms like sampled muzero to plan.

---

> ### Author Rebuttal · Authors · 2025-07-30
>
> Thank you for recognizing the novelty of our work. In what follows, we address the raised questions and weaknesses point-by-point.
>
> **W1: The paper seems to have a lot of technical details about UCB, linear bandits and regret analysis. This makes it a little difficult to read for practitioners / non-theory researchers. It also makes it less clear if the main contributions are theoretical / empirical. I do not work in multi-agent RL / multi-agent RL theory, so it is a bit difficult to gauge how strong the theoretical / empirical results are.**
>
> Our main contributions are two-fold: (i) The novel use of linear contextual bandit to enable factorization of multi-agent MCTS problems and the derivation of various guarantees; (ii) The design of MALinZero algorithm demonstrating significant reduction in joint action search. State-of-the-art MCTS algorithms such as MuZero and MAZero do not consider this type of factorization in terms of confidence bounds, making them not scalable as the number of agents increase. Empirical and the extensive experiments have shown consistent performance improvements over all baselines in benchmarks such as MatGame, SMAC and SMACv2.
>
> The algorithm design of MALinZero is based on solid theories. Specifically, we apply the contextual bandit theory [1] to design LinUCT which enables the low-dimensional representation of joint action and provides theoretical guarantee for regret analysis. Moreover, we propose a \\((1-\frac{1}{e})\\) approximation algorithm based on the submodular maximization theory [2] to select joint action. Additionally, to achieve efficient Backpropagation practically in MCTS we update the parameter according to the Sherman-Morrison formula [3], successfully reducing the computing complexity from \\(\mathcal O(n^2d^2)\\) to the linear order \\(\mathcal{O} (nd)\\) where \\(n\\) is the anget number and \\(d\\) is the space size of single action.
>
> **W2: How relevant do these algorithms remains when compute becomes orders of magnitude cheaper? Every agent can consider other agents as a part of the environment and use algorithms like sampled muzero to plan.**
>
> **Q3: What happens when compute becomes orders of magnitude cheaper? Do you think the low rank approximation will still achieve better performance?**
>
> First, we want to point out that MALinZero improves complexity from exponential to linear with respect to the number of agents. Consider a decision-making problem involving $n$ agents, each with $d$ actions. A standard approach (such as SOTA method MAZero) would need to consider sampling \\(\mathcal{O}(d^n)\\) possible joint actions (to estimate their confidence bounds). MALinZero considers a linearly factorized structure reducing this to sampling \\(\mathcal{O}(dn)\\) actions. Consider a problem with $n=10$ agents and $d=8$ actions. This is a reduction from $8^{10}$ to $8\times 10$, which goes beyond computation cost factor.
>
> We would also like to clarify that there are two different types of complexity involved here. One is sampling complexity (i.e., sampling the game environment with actions chosen) and the other is computational complexity. Sampling a physical environment (e.g., autonomous driving) has overhead/cost factors outside computation. By reducing the sampling overhead from exponential to linear, MALinZero achieves a huge saving that is not seperate from computation cost.
>
> The mentioned distributed training and distributed execution (DTDE) must learn with respect to changing behavior of other agents (which are also updated during training), when they are considered as part of an environment (that now becomes stochastic).
>
> **Q1: Do you need to learn a reward function new approximation for different latent states? How computationally efficient is this? How is data generated to learn this function?**
>
> In MALinZero, the reward function is trained with the representation function jointly so we do not need to learn a reward function for different latent states. This is the same method used in state-of-the-art MCTS solutions, such as MuZero and MAZero. It has been shown to be efficient even for very complex games such as go and chess.
>
> To provide more details, the input of reward function are the states \\((s^1_t,\dots,s^n_t)\\) and actions \\((a^1_t,\dots,a^n_t)\\) of all agents at time step \\(t\\) and the output is the reward \\(r_t\\) for taking the input action at the input state. The training data comes from the interaction of model with the environment, which is also a conventional strategy to get training data for online reinforcement learning. For better understanding, we can wrap the complex architecture and networks of MALinZero into a simple model-based algorithm since the training procedure remains the same. At the beginning of training, the untrained model interacts with the environment and then some trajectories including \\(s,a,o,r\\) are generated, where \\(s,a,o\\) is the state-action-obervation tuple and \\(r\\) is the reward given by the environment. When there are sufficient trajectories in the buffer, the algorithm could apply them to train the reward function and other networks as conventional reinforcement learning algorithms do. The learned model will interact with the environment to generate new data and these new data are further used to train the model.
>
>
> **Q2: At every-step during expansion, you need to run the linUCB algorithm to select an action, what additional computational and memory overheads does this cause?**
>
> This is a good questions. Yes, running the LinUCB algorithm will incur minor additional cost. However, the computation leverages a linear structure with sampling \\(\mathcal{O}(dn)\\) actions rather than the standard \\(\mathcal{O}(d^n)\\). Compared with this reduction from exponential to linear, the additional computation cost is negligible. Our evaluation shows that the computational cost is comparable to that of the MAZero [4] method. We will include additional explanations in the revised version.
>
> [1] Wang, Chih-Chun, Sanjeev R. Kulkarni, and H. Vincent Poor. "Bandit problems with side observations." IEEE Transactions on Automatic Control 50.3 (2005): 338-355.
>
> [2] Nemhauser, George L., Laurence A. Wolsey, and Marshall L. Fisher. "An analysis of approximations for maximizing submodular set functions—I." Mathematical programming 14.1 (1978): 265-294.
>
> [3]Sherman, Jack, and Winifred J. Morrison. "Adjustment of an inverse matrix corresponding to a change in one element of a given matrix." The Annals of Mathematical Statistics 21.1 (1950): 124-127.
>
> [4] Liu, Qihan, et al. "Efficient multi-agent reinforcement learning by planning." arXiv preprint arXiv:2405.11778 (2024).

---

> ### Comment · Area_Chair_kuEJ · 2025-08-05
>
> Thanks to the authors for their rebuttal! Since this paper got borderline reviews, a critical evaluation whether the rebuttal addresses the reviewer's questions and criticisms is very important. Could the reviewer please check whether the rebuttal left them still with open questions, whether their concerns are alleviated enough to raise their score, or just acknowledge that neither is the case?

---

> > ### Comment · Reviewer_oPgB · 2025-08-05
> > **Reply by reviewer**
> >
> > I thank the authors for their rebuttal. It answered my questions and helped clarify the contributions.
> >
> > I will maintain my positive assessment of the paper. I believe this paper should get accepted.

---

> > > ### Author Response · Authors · 2025-08-07
> > >
> > > Thanks for your feedback and support!

---

### Official Review · Reviewer_hYgq · 2025-07-03

**Clarity:** 3
**Significance:** 3
**Originality:** 3
**Rating:** 5
**Confidence:** 2

**Summary:**

This paper introduces MALinZero, a novel algorithm for multi-agent planning that incorporates low-dimensional representations of joint-action returns into Monte Carlo Tree Search (MCTS). The method models joint action values as a contextual linear bandit problem and derives a variant of Upper Confidence Bounds for trees, dubbed LinUCT, to guide efficient exploration. The paper presents regret analysis, derives an approximate algorithm to minimize it, an evaluates it over against existing baselines on MatGame, SMAC, and SMACv2.

**Questions:**

See above.

**Ethical Concerns:**

["NO or VERY MINOR ethics concerns only"]

**Final Justification:**

The authors have address my concerns, thus I have increased my score.

**Limitations:**

See above.

**Paper Formatting Concerns:**

None.

**Quality:**

3

**Strengths And Weaknesses:**

The paper is very well written, ideas and results are presented clearly, and related work is well covered.
I am not very familiar with the subject, though, so I may not be aware of missing references.

While the idea of low-dimensional decomposition has been explored in previous MARL work (the authors cite QMIX and VD), the idea of transforming joint action return estimation in MCTS into a contextual linear bandit problem seems novel to me, and useful especially in MA problems where the joint action space grows exponentially.

- How does MALinZero differs in terms of how it uses confidence bounds for exploration in contrast to value factorization? As someone not very familiar with the subject, I would appreciate a concise and clear conceptual comparison against baselines, and a recap of contributions at the end of the paper.

- How does MALinZero scale to larger problems? The problem of scalability is briefly mentioned in the related work section, but not addressed later. Is MALinZero better than baselines in this regard?

- How would MALinZero perform with different losses? E.g., Huber or exponential.

The authors also provide an extensive empirical evaluation, showing that MALinZero consistently outperforms all baselines across linear and nonlinear reward environments.

- All algorithms are evaluated over 3 seeds only. I am not familiar with the environments, but in classic RL 3 seeds are usually unacceptable. I strongly encourage the authors to run more seeds. Also, please indicate what the shaded are and the +/- in the table are (standard deviation? confidence interval?).

---

> ### Author Rebuttal · Authors · 2025-07-30
>
> Thank you very much for your positive recommendation of our work and your insightful comments. Following are our responses to your concerns.
>
> **Q1: How does MALinZero differs in terms of how it uses confidence bounds for exploration in contrast to value factorization? As someone not very familiar with the subject, I would appreciate a concise and clear conceptual comparison against baselines, and a recap of contributions at the end of the paper.**
>
> Value-function factorization methods in multi-agent reinforcement learning (such as VDN, PAC, Qatten, DOP, FOP, PAC) do not explicitly model the uncertainty of the learned Q-value, which are resulted from limited samples during exploration-exploitation. They often choose optimal actions based on the expected Q-values -- which are obtained following a factorized structure -- and then rely on heuristics or introduce regularization terms to encourage exploration. Examples include $\epsilon$-greedy (taking the optimal action \\(a_t=\arg\max_a Q_i(s_i,a)\\) with the probability of \\((1-\varepsilon_t)\\) and a random action with probability \\(\varepsilon_t\\)); soft-actor critic to an entropy term encouraging exploration; and introducing intrinsic reward terms to explore unseen states/actions.
>
> In contrast, MALinZero models the Q-values from limited samples as distributions (which are estimated using limited samples) and derive mathematical guarantees on the potential return of each joint action through confidence bounds. It considers a multi-armed bandit formulation to provide a formal, rigorous framework of considering the uncertainty in exploration-exploitation. Compared with previously mentioned algorithms, the exploration using confidence bound can focus on the action with both high return and high potential automatically, instead of trying each action with the same probability \\(\varepsilon\\).
> These MCTS based solutions are known to be significantly more sample efficient than multi-agent reinforcement learning, as demonstrated by single-agent MCTS algorithms like MuZero and MAZero.
>
> Our paper proposes a factorized confidence bound (rather than a factorized Q value) to guide exploration-exploitatio in MCTS, when multiple agents are involved. Intuitively, consider a linear factorization of $Q$ into $Q_1, Q_2, ... Q_n$. Estimating the mean of $Q$ only requires a sum over the mean of each $Q_i$. But a confidence bound on the joint $Q$ must evaluate the probability $P(Q-\bar{Q})<\epsilon$ with respect to the joint distribution of $Q_i$s using limited samples. Moreover, the adaptive and dynamic update of confidence bound is more suitable for long-horizon decision-making, especially with the help of the tree structure while \\(\varepsilon\\)-greedy deals with single-step stochastic exploration. In conclusion, we apply the LinUCT to facilitate the exploration and exploitation in the low-dimensional space, which deals with uncertainties more efficiently and effectively with faster convergence compared with baselines. We will add the comparison and recap of contributions in a further version of our paper.
>
> **Q2: How does MALinZero scale to larger problems? The problem of scalability is briefly mentioned in the related work section, but not addressed later. Is MALinZero better than baselines in this regard?**
>
> The goal of MALinZero is to improve scalability by introducing a linear structure for efficient evaluation of the confidence bound. Consider a decision-making problem involving $n$ agents, each with $d$ actions. A standard approach (such as SOTA method MAZero) would need to consider sampling \\(\mathcal{O}(d^n)\\) possible joint actions (to estimate their confidence bounds). MALinZero considers a linearly factorized structure reducing this to sampling \\(\mathcal{O}(dn)\\) actions. This is a significant reduction especially in systems with large number $n$ of agents.
>
> As a result, MALinZero only samples fixed number of child nodes for each node in the Expansion stage instead of traversing the exponentially large action space to search candidate actions. Moreover, we leverage the Sherman-Morrison formula [1] to reduce the computing complexity of updating parameter \\(\theta\in\mathbb{R}^{nd}\\) form \\(\mathcal{O}(n^2d^2)\\) to \\(\mathcal{O}(nd)\\). Therefore, the computational cost of each step of MALinZero is only linearly related to the product of the number of agents \\(n\\) and the number of actions \\(d\\) of each agent, rather than an exponential relationship; compared with MAZero, its scalability does not depend on the explosive growth of the number of joint actions, and it actually performs better in scenarios with large \\(n\\) and \\(d\\). We note that there are some additional cost involved in Dynamic Node Generation (DNG) using MALinZero, but it is insignificant compared with the substantial reduction of search space.
>
> **Q3: How would MALinZero perform with different losses? E.g., Huber or exponential.**
>
> We choose our current loss to be consistent with existing search-based reinforcement learning algorithms both for single-agent problems [2] [3] and multi-agent problems [4]. Since the loss is proven useful and powerful in model-based reinforcement learning algorithms equipped with MCTS, we also adopt this loss. This will ensure a fair comparison with the baselines.  In this work, we want to emphasize the novelty in terms of low-dimensional representation, LinUCT, and submodular maximization-based joint action selection instead of designing a new loss.
>
> **Q4: All algorithms are evaluated over 3 seeds only. I am not familiar with the environments, but in classic RL 3 seeds are usually unacceptable. I strongly encourage the authors to run more seeds. Also, please indicate what the shaded are and the +/- in the table are (standard deviation? confidence interval?).**
>
> Thanks for pointing this out. To address the concern, we further evaluate algorithms on two MatGame environments (6,8,Linear) and (6,8,Non Linear) over 5 random seeds and report the results here.
> | Agent | Action | Type       | Steps | MAZero       | MAZero-NP     | MA‑AlphaZero | MAPPO         | QMIX          | **MALinZero** |
> |:-----:|:------:|:----------:|:-----:|:------------:|:-------------:|:------------:|:-------------:|:-------------:|:-------------:|
> | 6     | 8      | Linear     | 1000  | 393.2±9.5    | 387.6±9.8     | 389.9±8.7    | 391.8±10.1    | 384.9±10.7    | **395.8±9.1** |
> | 6     | 8      | Linear     | 2000  | 432.3±6.9    | 426.1±9.2     | 433.1±9.4    | 430.2±8.9     | 431.2±7.3     | **438.2±6.3** |
> | 6     | 8      | Non‑Linear | 1000  | 399.1±12.9   | 390.4±10.1    | 392.3±11.9   | 389.1±13.7    | 391.3±12.8    | **407.2±7.3** |
> | 6     | 8      | Non‑Linear | 2000  | 442.3±11.8   | 427.3±8.4     | 425.9±7.8    | 428.7±8.9     | 433.2±8.1     | **449.8±12.1**|
>
> The shaded area and the \(+/-\) in this paper refer to one stand deviation. At the beginning of conducting experiments, we also evaluated all algorithms over more seeds. We found relatively consistent results over different seeds.
>
>
>
> [1] Sherman, Jack, and Winifred J. Morrison. "Adjustment of an inverse matrix corresponding to a change in one element of a given matrix." The Annals of Mathematical Statistics 21.1 (1950): 124-127.
>
> [2] Schrittwieser, Julian, et al. "Mastering atari, go, chess and shogi by planning with a learned model." Nature 588.7839 (2020): 604-609.
>
> [3] Hubert, Thomas, et al. "Learning and planning in complex action spaces." International Conference on Machine Learning. PMLR, 2021.
>
> [4] Liu, Qihan, et al. "Efficient multi-agent reinforcement learning by planning." arXiv preprint arXiv:2405.11778 (2024).

---

> > ### Comment · Reviewer_hYgq · 2025-08-02
> >
> > Thanks for the reply, it clarifies all my question.
> > Just one follow-up. How does it relate to distributional Q-learning? Like quantile DQN? It's one thing that came to my mind while reading your response to Q1.

---

> > > ### Author Response · Authors · 2025-08-02
> > >
> > > Thanks for your feedback!
> > >
> > > MALinZero and distributional Q-learning both move beyond a single “point estimate” of $Q$-value, but they do so in very different ways. In quantile DQN (and other distributional methods) we learn a full approximate return distribution $Z(s,a)$—e.g. by estimating its quantiles—mainly to get a richer value representation. MALinZero, by contrast, does not try to approximate the entire distribution of returns. Instead, it uses a linear bandit model over features to derive analytic upper-confidence bounds on the joint $Q$–value.

---

### Note · Authors · 2025-08-12

We thank all reviewers for their constructive feedback and final positive comments on our work! It is our pleasure to address nearly all the concerns that have been raised, and two reviewers raised their scores and the other two maintained the positive ratings. We also appreciate the contributions to organizing the review of the Area Chair.

Concerns mainly focus on three aspects: 1) Reviewers are curious about how MALinZero scales to larger problems and performs with fewer computing resources. 2) More detailed explanations for the algorithm design and concepts are required to understand MALinZero better, such as how the loss function\\(f\\) works. 3) A more elegant paper organization is encouraged to improve the clarity. We analyzed the computing complexity of MALinZero and state it is \\(\mathcal{O}(nd)\\) where $n$ is the number of agents and $d$ is the action space size, which scales linearly with the system size. We have provided additional explanations for the mechanism and design of MALinZero in the rebuttal with examples, and we will add more details about the design and implementation in the next version of the paper. Detailed contents, such as implementation details and proof sketch, will be added to the paper properly to improve the overall writing quality.

We have added experiments on more random seeds to show the consistent performance of the proposed algorithm. Additionally, we conducted extra experiments to demonstrate how the visit counts are distributed and help understand the effectiveness of MALinZero.

---

### Decision · Program_Chairs · 2025-09-17

**Decision:**

Accept (poster)

**Comment:**

This paper is right at the threshold between acceptance and rejection. It adapts a known method from MARL for an extension of MCTS to MARL. Reviewers found it well written, but too many details have been moved into the appendix for full comprehension. However, the paper's highlight are the UCB regret bounds for the linearized representation of the joint action space that make MCTS planning efficient. Their approach comes at the price of all agent needing access to all others' observations, which is much more restrictive than model-free approaches to solve Dec-POMDP like QMIX. Nonetheless, the resulting AlphaZero style algorithm performs very well on the SMAC benchmark against other model-based baselines with the same assumptions. On the more conceptual MatGame evaluations other MARL planning algorithms (in particular MAZero) do not perform significantly worse, though, but only the proposed method is printed in bold, which misleads reader. This needs to be rectified.

Reviewers are mostly in favor of acceptance, as the paper presents a theoretically justified method that allows MARL planning with MCTS. On the other hand, the current version is hard to follow without reading the appendix, which should never be a requirement. The authors promise to fix this, but this is easy to promise and hard to do.

I do follow the reviewers and recommend acceptance as well, but this remains a threshold decision. I would therefore not mind if this paper gets bumped down due to limited conference resources.